# The importance of DNAPKcs for blunt DNA end joining is magnified when XLF is weakened

Metztli Cisneros-Aguirre[1,2], Felicia Wednesday Lopezcolorado[1], Linda Jillianne Tsai[1,2], Ragini Bhargava[1,2,3] & Jeremy M. Stark [1,2✉]

Canonical non-homologous end joining (C-NHEJ) factors can assemble into a long-range (LR) complex with DNA ends relatively far apart that contains DNAPKcs, XLF, XRCC4, LIG4, and the KU heterodimer and a short-range (SR) complex lacking DNAPKcs that has the ends positioned for ligation. Since the SR complex can form de novo, the role of the LR complex (i.e., DNAPKcs) for chromosomal EJ is unclear. We have examined EJ of chromosomal blunt DNA double-strand breaks (DSBs), and found that DNAPKcs is significantly less important than XLF for such EJ. However, weakening XLF via disrupting interaction interfaces causes a marked requirement for DNAPKcs, its kinase activity, and its ABCDE-cluster autophosphorylation sites for blunt DSB EJ. In contrast, other aspects of genome maintenance are sensitive to DNAPKcs kinase inhibition in a manner that is not further enhanced by XLF loss (i.e., suppression of homology-directed repair and structural variants, and IR-resistance). We suggest that DNAPKcs is required to position a weakened XLF in an LR complex that can transition into a functional SR complex for blunt DSB EJ, but also has distinct functions for other aspects of genome maintenance.

[1] Department of Cancer Genetics and Epigenetics, Beckman Research Institute of the City of Hope, 1500 E Duarte Rd, Duarte CA 91010, USA. [2] Irell and Manella Graduate School of Biological Sciences, Beckman Research Institute of the City of Hope, 1500 E Duarte Rd, Duarte CA 91010, USA. [3] Present address: Department of Pharmacology and Chemical Biology, UPMC Hillman Cancer Center, University of Pittsburgh, Pittsburgh, PA, USA. ✉email: jstark@coh.org

Repair of DNA double-strand breaks (DSBs) is critical for genome stability, and is a central aspect of genome editing[1–3]. DNA DSB repair is also a mechanism of cancer cell therapeutic resistance to clastogenic agents, such as ionizing radiation (IR)[4]. Canonical non-homologous end joining (C-NHEJ) is a major DSB repair pathway that functions throughout the cell cycle, and involves the factors KU70/80 (KU), XRCC4, DNA Ligase 4 (LIG4), XLF, and DNAPKcs[5,6]. Accordingly, characterizing the mechanism of C-NHEJ provides insight into genome maintenance, tumor resistance to clastogenic therapeutics, and gene editing.

A central aspect of C-NHEJ is synapsis of the two DNA ends to enable efficient ligation. Single-molecule studies in *Xenopus* extracts identified both a long-range (LR) synapsis interaction between labeled DNA molecules without fluorescence resonance energy transfer (FRET), and a short-range (SR) synapsis interaction with detection of FRET between the labels[7]. Recent cryoelectron microscopy (cryo-EM) structures have identified complexes that can mediate such LR and SR interactions[8,9]. For one, DNAPKcs and KU bound to each DNA end are able to support an LR interaction[9], although DNAPKcs and KU can also form stable complexes, including in the active kinase form, on individual DNA ends[10]. Apart from these complexes with DNAPKcs and KU alone, an additional complex has been identified that also includes XRCC4, XLF, and LIG4, which we will refer to as the LR complex[8,9]. For this LR complex, each DNA end is bound to one molecule of DNAPKcs, KU, XRCC4, and LIG4[8,9]. In addition, a single XLF homodimer appears to bridge the two DNA ends through interactions of each XLF monomer with XRCC4 and KU[8,9]. As mentioned above, the DNA ends in this complex are not close enough to facilitate ligation. However, cryo-EM studies have also identified an SR complex with DNA ends positioned to enable ligation[8]. This SR complex has commonalities with the LR complex, in that it also contains a single XLF homodimer bridging two DNA ends that are each bound to a molecule of XRCC4, KU, and LIG4[8]. However, this SR complex does not contain DNAPKcs[8].

Whether the LR complex is a prerequisite to form a functional SR complex in cells remains controversial. From studies in *Xenopus* extracts, DNAPKcs is important not only to form the LR interaction, but also for the transition to the SR interaction, in a manner dependent on DNAPKcs kinase activity[7]. However, using purified proteins the SR complex appears to form without DNAPKcs. Namely, as mentioned above, the cryo-EM structure of the SR complex was generated without DNAPKcs[8]. Similarly, also with purified proteins, close end synapsis interactions have been shown to form with KU, XLF, XRCC4 and LIG4 without requiring DNAPKcs[11,12]. These studies raise the notion that under some circumstances the SR complex can form de novo, i.e., without prior formation of the LR complex that contains DNAPKcs.

Furthermore, the requirement for the DNAPKcs LR complex for chromosomal DSB EJ has remained unclear, particularly outside the context of V(D)J recombination. V(D)J recombination is a programmed rearrangement that involves joining of both hairpin coding ends and signal ends[13,14]. For V(D)J recombination, DNAPKcs is important to recruit the Artemis nuclease for coding EJ, promotes both coding and signal EJ of plasmid substrates, but is not essential for chromosomal signal EJ in mouse lymphocytes unless paired with the loss of other factors (i.e., XLF or the ATM kinase)[13,15–18]. Separate from this programmed C-NHEJ event, the role of DNAPKcs on chromosomal EJ is poorly understood. Indeed, recent studies have suggested that DNAPKcs largely functions during genome maintenance to regulate cellular senescence and chromatin decondensation, as well as playing a role in rRNA processing[19–21]. This gap in knowledge on the role of DNAPKcs for chromosomal EJ is partly due to

limitations of DSB repair assays that measure chromosomal rearrangements and/or insertion/deletion mutations (indels) that can also be mediated by the Alternative EJ pathway[14,22–25]. However recent studies using the RNA guided nuclease Cas9, which largely induces blunt DSBs[26–28], have found that EJ of two Cas9 blunt DSBs without indels (No Indel EJ) is markedly dependent on C-NHEJ (i.e., XRCC4, LIG4, XLF, and KU)[29–31]. Thus, we have sought to define the role of DNAPKcs for EJ of blunt chromosomal DSBs, as well as other DSB repair outcomes. In particular, we have used combination mutants to examine possible partial redundancies between XLF and DNAPKcs.

## Results

**DNAPKcs is less important for No Indel EJ vs. XLF and XRCC4, but becomes essential in combination with an XLF mutant (K160D).** We have sought to understand the role of DNAPKcs on EJ outcomes, with a focus on EJ of blunt chromosomal DSBs. This type of EJ does not involve an annealing intermediate that could facilitate Alternative EJ, and therefore likely requires stable DSB end bridging via the C-NHEJ complex to facilitate efficient ligation. To begin with, we used the EJ7-GFP reporter (Fig. 1a) that measures EJ of blunt DNA DSBs without indels (No Indel EJ). For these experiments, EJ7-GFP was chromosomally integrated into human HEK293 cells, mouse embryonic stem cells (mESCs), and human U2OS cells. This reporter contains a green fluorescent protein (GFP) sequence interrupted by a 46 bp spacer at a glycine codon critical for GFP fluorescence (Fig. 1a). We induced two blunt DSBs at the edges of the 46 bp spacer using Cas9 and two single guide RNAs (sgRNAs), which are introduced by transfection of expression plasmids. If the distal blunt DSB ends are repaired via No Indel EJ, this repair event leads to GFP expressing cells that are detected via flow cytometry. Using this assay, No Indel EJ was shown to be markedly promoted by the C-NHEJ factors XRCC4, XLF, and KU70[31], and we sought here to examine the influence of DNAPKcs (*PRKDC* gene) on such EJ.

Beginning with HEK293 cells with the EJ7-GFP reporter, we examined several knockout (KO) cell lines and found that there was a significant decrease in No Indel EJ in *PRKDC-KO*, *XLF-KO*, and *XRCC4-KO* cells compared to the parental HEK293 cells (Fig. 1b). However, for *PRKDC-KO* cells, the No Indel EJ frequency was significantly higher compared to *XLF-KO* cells, which itself was significantly higher compared to *XRCC4-KO* cells (Fig. 1b). We found similar results with U2OS cells, except DNAPKcs loss caused a more modest reduction in No Indel EJ, as compared to HEK293 (Supplementary Fig. 1a). Notably, the relative levels of DNAPKcs appear lower in U2OS vs. HEK293 (Supplementary Fig. 1b). Finally, using HEK293 cells, we performed experiments with expression vectors for XRCC4, XLF, and DNAPKcs in the respective mutant cell lines, finding that XLF and XRCC4 expression restored No Indel EJ levels, whereas DNAPKcs expression caused only a modest increase (Fig. 1b). However, we note that the XLF and XRCC4 expression vectors caused overexpression of these proteins, whereas the DNAPKcs expression vector did not restore endogenous levels (Fig. 1b). Altogether, these findings indicate that DNAPKcs is relatively less important for No Indel EJ compared to XLF and XRCC4 in two human cell lines.

We then posited that the residual No Indel EJ in both the *PRKDC-KO* and *XLF-KO* cell lines is due to partial functional redundancies between DNAPKcs and the XLF homodimer (Fig. 1c) to mediate DNA end bridging[8,9,32,33]. To begin to test this hypothesis, we examined No Indel EJ in *XLF-KO/PRKDC-KO* double-mutant HEK293 cells, and found that No Indel EJ was ablated compared to the *XLF-KO* cells (Fig. 1d).

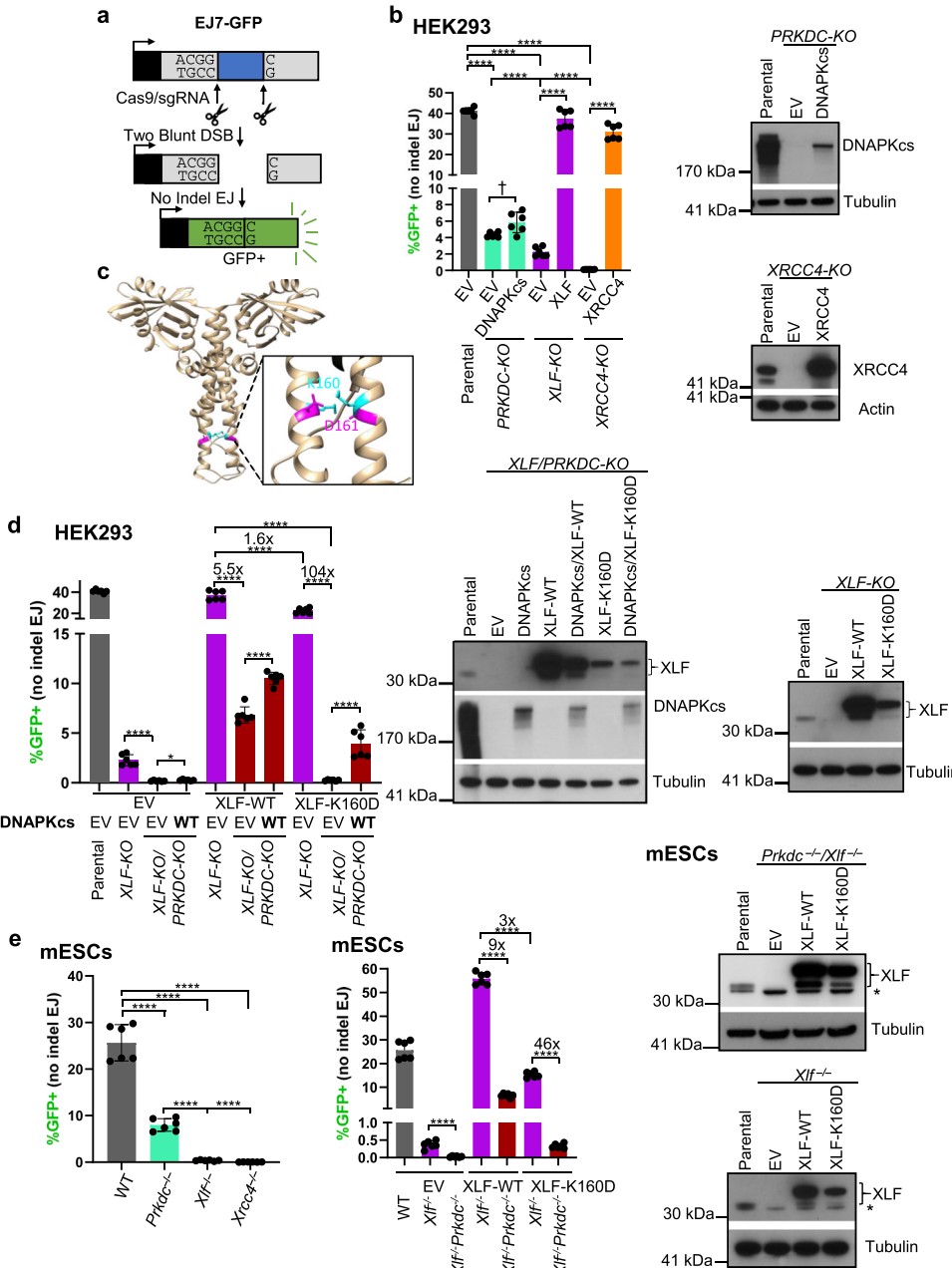

**Fig. 1 DNAPKcs is less important for No Indel EJ vs. XLF and XRCC4, but becomes essential in combination with an XLF mutant (K160D). a** Shown is the EJ7-GFP reporter for No Indel EJ (not to scale), which is chromosomally integrated using the Flp-FRT system in HEK293 and U2OS cells, and targeting to the *Pim1* locus of mESCs. Cells are transfected with expression vectors for Cas9 and the two sgRNAs, with complementing vector(s) or empty vector (EV), and GFP frequencies are normalized to transfection efficiency with parallel GFP transfections. **b** DNAPKcs is less important for No Indel EJ compared to XLF and XRCC4 in HEK293 cells. $n = 6$ biologically independent transfections. Statistics with unpaired two-tailed $t$ test using Holm–Sidak correction. ****$P < 0.0001$, †$P = 0.0307$ but not significant after correction (unadjusted $P$-value). Immunoblots show levels of DNAPKcs and XRCC4. **c** A reported structure of the XLF homodimer (aa 1–227, Protein Data Bank 2R9A), with the K160/D161 salt-bridge highlighted. **d** DNAPKcs is required for No Indel EJ in HEK293 cells with XLF-K160D. $n = 6$ biologically independent transfections. Statistics as in (**b**), #x represents fold effect. ****$P < 0.0001$, *$P = 0.0305$. Immunoblots show levels of DNAPKcs, XLF-WT, and XLF-K160D. **e** The influence of DNAPKcs on No Indel EJ is conserved in mESCs. $n = 6$ biologically independent transfections. Statistics as in (**b**). ****$P < 0.0001$. Immunoblots show levels of XLF-WT and XLF-K160D. Data are represented as mean values ± SD.

Next, we tested whether weakening XLF binding interfaces that could mediate DSB end bridging might reveal a critical role for DNAPKcs in No Indel EJ. To begin with, given that a single XLF homodimer has been shown in recent cryo-EM structures to bridge DNA ends via interactions with other proteins in the C-NHEJ complex[8,9], we tested whether a mutation in the XLF coiled-coil dimerization domain may reveal a greater role for DNAPKcs. Specifically, we mutated the K160 residue, which is within the coiled-coil dimerization interface and is predicted to form a salt bridge with the D161 residue on the other monomer (Fig. 1c)[31,33,34]. Thus, a XLF-K160D mutant would be predicted to disrupt this salt bridge, and indeed has been shown to increase the coiled-coil interface distance within the XLF homodimer, based on molecular dynamics modeling[31,33,34]. Thus, we posited

that the XLF-K160D mutant would cause a greater dependency on DNAPKcs for No Indel EJ.

To test this hypothesis, we first compared the ability for transient expression of the control XLF-WT to promote No Indel EJ in the presence or absence of DNAPKcs (i.e., in *XLF-KO* vs. *PRKCD-KO/XLF-KO*), and found that loss of DNAPKcs caused a 5.5-fold reduction in No Indel EJ (Fig. 1d). We then compared XLF-WT vs. XLF-K160D in the presence of DNAPKcs (i.e., *XLF-KO*), and found that XLF-K160D caused a modest decrease in No Indel EJ vs. XLF-WT (1.6-fold) (Fig. 1d). Although, we note that XLF-K160D is expressed at a lower level than XLF-WT, albeit at a higher level than endogenous XLF, which may reflect intrinsic instability of an XLF mutant that disrupts the dimer interface. In any case, the XLF-K160D mutation has only a modest effect in the presence of DNAPKcs. However, when we examined the double mutant (XLF-K160D in *XLF-KO/PRKDC-KO* cells), we found a marked loss of No Indel EJ (Fig. 1d, 104-fold decrease vs. *XLF-KO* cells expressing XLF-K160D). Notably, we also found that transient expression of DNAPKcs in the double-mutant (XLF-K160D in *XLF-KO/PRKDC-KO*) was able to substantially complement No Indel EJ (Fig. 1d). Although, the levels of such EJ did not recover to the levels of the single mutant (XLF-K160D in *XLF-KO*) likely because the expression vector was unable to restore endogenous levels of DNAPKcs (Fig. 1d), as described above (Fig. 1b). We also tested the effect of loss of DNAPKcs on No Indel EJ in mESCs with XLF-WT vs. XLF-K160D, and found similar results (Fig. 1e). Although for the mESC experiments, we found that while the 3xFLAG-XLF expression vectors predominantly cause expression of protein consistent with the full-length product, a minor product with a lower molecular weight is also detected (Fig. 1e). Altogether, these findings indicate that the XLF-K160D mutant causes a marked requirement for DNAPKcs for No Indel EJ.

**Inhibiting DNAPKcs kinase activity suppresses No Indel EJ to a greater degree when combined with XLF-K160D.** We then performed similar experiments with inhibition of DNAPKcs kinase activity, which appears to block its dissociation from DNA ends[7,8,35–37]. To begin with, we interrogated the effect of inhibiting DNAPKcs kinase activity with the small molecule inhibitor M3814 (i.e., Nedisertib)[38,39], and found a concentration dependent decrease in No Indel EJ events in HEK293 (Fig. 2a). Notably, we did not see a concentration dependent decrease in phosphorylation of DNAPKcs at S2056 via immunoblotting, suggesting the EJ7-GFP reporter for No Indel EJ is more sensitive to the concentration of M3814 (Fig. 2b). We then examined the effect of M3814 on No Indel EJ in *XLF-KO* cells expressing XLF-WT and found similar results to the parental HEK293 cells (Fig. 2a). However, combining XLF-K160D and M3814 treatment caused a marked loss of No Indel EJ (Fig. 2a). For example, 500 nM M3814 treatment caused a 3.8-fold decrease with XLF-WT, whereas addition of 500 nM M3814 to XLF-K160D caused a 31-fold decrease (Fig. 2a). We also tested the combination of XLF-K160D and M3814 on No Indel EJ in *XLF-KO* U2OS cells and again found a marked defect (Fig. 2c, Supplementary Fig. 1c). These findings indicate that XLF-K160D causes an enhanced requirement for DNAPKcs kinase activity for No Indel EJ.

We performed additional controls for these experiments, using the HEK293 cell lines. For one, we confirmed that M3814 treatment had no effect on No Indel EJ in the absence of DNAPKcs (Supplementary Fig. 1d). We also examined the effect of M3814 treatment on signaling events mediated by other kinases related to DNAPKcs: ATR (CHK1-S345p), ATM (ATM-S1981p), and MTOR (ribosomal protein S6-S235/236p)[40–42]. We found that these phosphorylation events were not obviously

affected at the doses used in the above experiments (i.e., 250 nM, 500 nM, and 1000 nM, Supplementary Fig 2a), which is consistent with the notion that M3814 is a specific inhibitor of DNAPKcs[38,39], and supports the use of the intermediate dose of 500 nM for the below experiments. Finally, we also examined cellular localization of XLF-WT vs. XLF-K160D by immunofluorescence, and found similar staining patterns (i.e., staining in both the nucleus and cytoplasm, Supplementary Fig. 2b).

We also considered that altering the levels of XLF in HEK293 cells may affect the relative requirement for DNAPKcs for No Indel EJ. We first tested shRNA depletion of XLF[43], which we found did not affect the frequency of No Indel EJ with or without M3814 (Supplementary Fig. 3a). In contrast, using *PRKDC-KO* cells, shRNA depletion of XLF caused a marked (20-fold) reduction in No Indel EJ (Supplementary Fig. 3b). We then tested overexpression of XLF, which did not affect the frequency of No Indel EJ with or without M3814 (Supplementary Fig. 3c). In contrast, with *PRKDC-KO* cells, overexpression of XLF caused a significant increase in No Indel EJ (Supplementary Fig. 3c). For comparison, we tested the effects of expressing XLF-K160D in parental and *PRKDC-KO* cells, and found no effect on No Indel EJ (Supplementary Fig. 3c). Finally, overexpression of XRCC4 also had no effect No Indel EJ (Supplementary Fig. 3c). In summary, only in the circumstance of DNAPKcs loss do we observe effects of the relative levels of XLF on No Indel EJ.

**Inhibiting DNAPKcs kinase activity affects EJ junction patterns to a greater degree when combined with XLF-K160D.** We next interrogated our above hypothesis using a distinct approach that enables measurement of diverse EJ outcomes: the *GAPDH-CD4* rearrangement assay that uses endogenous genes in human chromosome 12[29]. This assay uses two sgRNAs to target Cas9 DSBs after the promoter of both the *GAPDH* and *CD4* genes, such that if there is a deletion rearrangement between the two DSBs (i.e., a GAPDH-CD4 rearrangement), *CD4* is expressed from the *GAPDH* promoter (Fig. 3a). These *CD4+* cells are detected by immunostaining, and following isolation by fluorescence-activated cell sorting (FACS), the GAPDH-CD4 rearrangement junctions are amplified and analyzed by deep sequencing.

To begin with, we examined the overall GAPDH-CD4 rearrangement frequency (i.e., percentage of *CD4+*) in various HEK293 cell conditions. To examine DNAPKcs kinase inhibitor treatment, we used the concentration of 500 nM M3814, which had intermediate effects on the EJ7-GFP assay, compared to 250 nM and 1000 nM (Fig. 2a). We found that such M3814 treatment caused a modest but significant decrease in *CD4+* cells (Fig. 3b). The *PRKDC-KO* and *XLF-KO* cells also showed a decrease in *CD4+* cells compared to parental cells, although the frequency for *PRKDC-KO* was higher than *XLF-KO* cells, indicating that XLF promotes this rearrangement more so than DNAPKcs. Next, we examined combinations of M3814 treatment with *XLF-KO* cells with and without expression of XLF-WT and XLF-K160D. We found that M3814 had no effect on *CD4+* frequencies for either *XLF-KO* cells or *XLF-KO* cells expressing XLF-WT (Fig. 3c). However, for *XLF-KO* cells expressing XLF-K160D, M3814 treatment caused a significant decrease in the *CD4+* cells, indicating DNAPKcs kinase activity is important for the GAPDH-CD4 rearrangement with XLF-K160D.

We then examined the frequency of distinct rearrangement junction types for each of the above conditions using amplicon deep sequencing of *CD4+* cells. Specifically, we classified the junctions as No Indel EJ (precise joining of the distal blunt Cas9 DSBs), deletions (loss of nucleotides), insertions (gain of nucleotides), or complex indel (combined loss/gain of nucleotides). We determined the frequency of each of these junction types based on

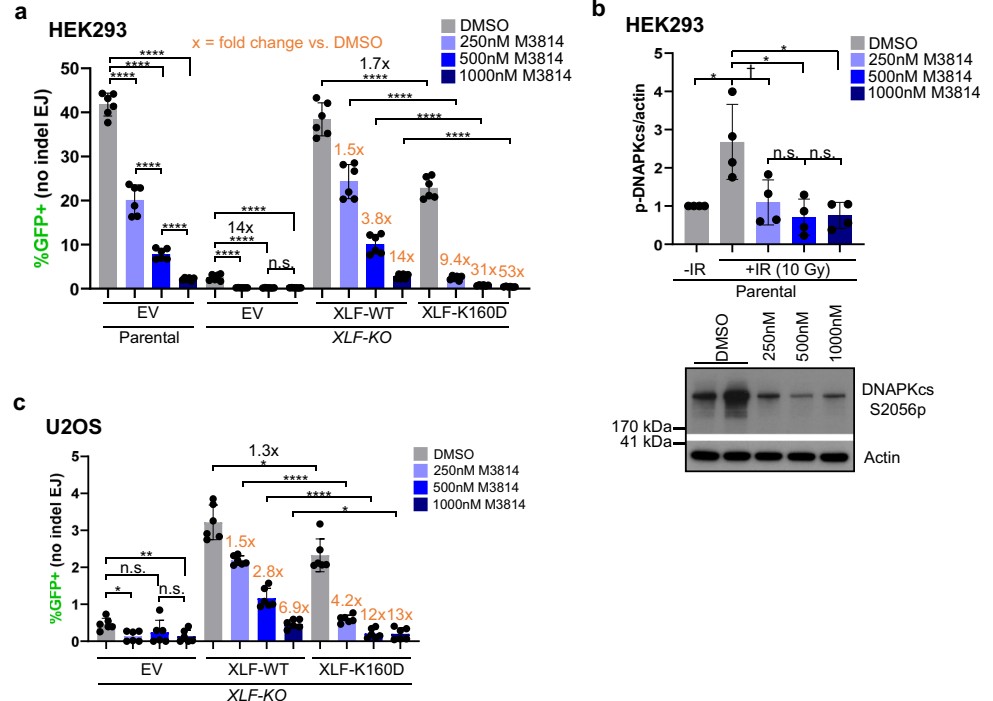

**Fig. 2 Inhibiting DNAPKcs kinase activity suppresses No Indel EJ to a greater degree when combined with XLF-K160D. a** Effect of DNAPKcs kinase inhibitor M3814 on No Indel EJ in HEK293 cells with XLF-WT, XLF-K160D, and without XLF. Control cells treated with DMSO (the vehicle for M3814). $n = 6$ biologically independent transfections. Statistics with unpaired two-tailed $t$ test using Holm–Sidak correction. ****$P < 0.0001$, n.s.= not significant. Fold-effects are shown to facilitate comparisons of the effect of each M3814 treatment on XLF-WT vs. K160D. **b** M3814 decreases ionizing radiation (IR)-induced phosphorylation of DNAPKcs at S2056 (S2056p). Shown are levels of DNAPKcs-S2056p normalized to Actin, and a representative immunoblot. $n = 4$ biologically independent wells of seeded cells. Statistics as in (**a**). DMSO −IR vs + IR *$P = 0.0142$, DMSO + IR vs. 500 nM M3814 + IR *$P = 0.0497$, DMSO + IR vs. 1000 nM M3814 + IR *$P = 0.0497$, DMSO + IR vs. 250 nM M3814 + IR †$P = 0.0328$ but not significant after correction (unadjusted $P$-value), n.s.= not significant. **c** The influence of DNAPKcs kinase inhibition with XLF-K160D is similar in U2OS cells. $n = 6$ biologically independent transfections. Statistics as in (**a**). ****$P < 0.0001$, EV DMSO vs. EV 1000 nM M3814 **$P = 0.00311$, EV DMSO vs. EV 250 nM M3814 *$P = 0.0116$, XLF-WT DMSO vs. XLF-K160D DMSO *$P = 0.042$, XLF-WT 1000 nM M3814 vs. XLF-K160D 1000 nM *$P = 0.0169$, n.s.= not significant. Data are represented as mean values ± SD.

read counts from *CD4*+ sorted cells, for three independent transfections and sorted samples per condition. Accordingly, the statistical analysis is based on the frequency for each junction type from the triplicate samples (i.e., $n = 3$).

From this analysis, we found that loss of DNAPKcs, as well as M3814 treatment caused a significant decrease in No Indel EJ, but the fold-effects were relatively modest (Fig. 3d). In contrast, *XLF-KO* cells showed a marked drop in No Indel EJ that was significantly lower than the *PRKDC-KO* cells (Fig. 3d). These findings are similar to the results with the EJ7-GFP reporter (Fig. 1). Interestingly, the influence of DNAPKcs and XLF on insertions was similar to the effect on No Indel EJ (Fig. 3d). However, we observed the converse effect on deletion mutations. Namely, *PRKDC-KO* cells, M3814 treatment of parental cells, and *XLF-KO* cells each showed a significant increase in the frequency of deletions, although the effect of XLF loss was greater than that of DNAPKcs loss (Fig. 3d). Finally, complex indel junctions were relatively infrequent for all of the conditions (Fig. 3d).

We then examined junction types in *XLF-KO* cells expressing XLF-WT or XLF-K160D, each with and without M3814 treatment, using 500 nM as above (Fig. 3e). For the No Indel EJ and insertion junction types, cells with XLF-K160D showed a modest decrease (1.3-fold and 1.6-fold, respectively) compared to XLF-WT, which was similar to the effects of M3814 treatment with XLF-WT (both 1.5-fold, Fig. 3e). Conversely, deletion junctions were increased (1.7-fold for XLF-K160D, 1.8-fold for M3814, Fig. 3e). However, M3814 treatment with XLF-K160D

caused a marked decrease in No Indel EJ (6.2-fold), a similar decrease in insertions (10-fold), and a converse increase in deletions (1.9-fold, Fig. 3e). In contrast, complex indel junctions were infrequent for all conditions (Fig. 3e). These findings indicate that combining XLF-K160D with DNAPKcs kinase inhibition causes a synergistic decrease in both No Indel EJ and insertions, along with a converse increase in deletions.

**The influence of DNAPKcs and XLF on the sizes of deletions and insertions at EJ junctions, and on a +1 insertion EJ event likely caused by staggered Cas9 DSBs.** We next evaluated various features of deletions and insertions for these EJ events. We began with examining deletion size (1–5, 6–10, 11–15, 16–20, >20 nts). From this analysis, we found that XLF loss and M3814 treatment caused a significant reduction in short deletions (1–5, 6–10 nts), and a converse increase in 16–20 nt deletions (Fig. 4a). Interestingly, this effect of M3814 treatment on deletion size was similar both in the absence of XLF, and in cells expressing XLF-K160D (Fig. 4b). With *PRKDC-KO* cells, the pattern was somewhat different, in that loss of DNAPKcs caused a reduction in 6–10 and 11–15 nt deletions, and a modest increase in 1–5 nt deletions (Fig. 4a). Altogether, these findings indicate that M3814 treatment causes a substantial shift to relatively large deletions in a manner that is not dependent on XLF.

Next, given that insertions and No Indel EJ showed similar genetic requirements in the above junction analysis, we further

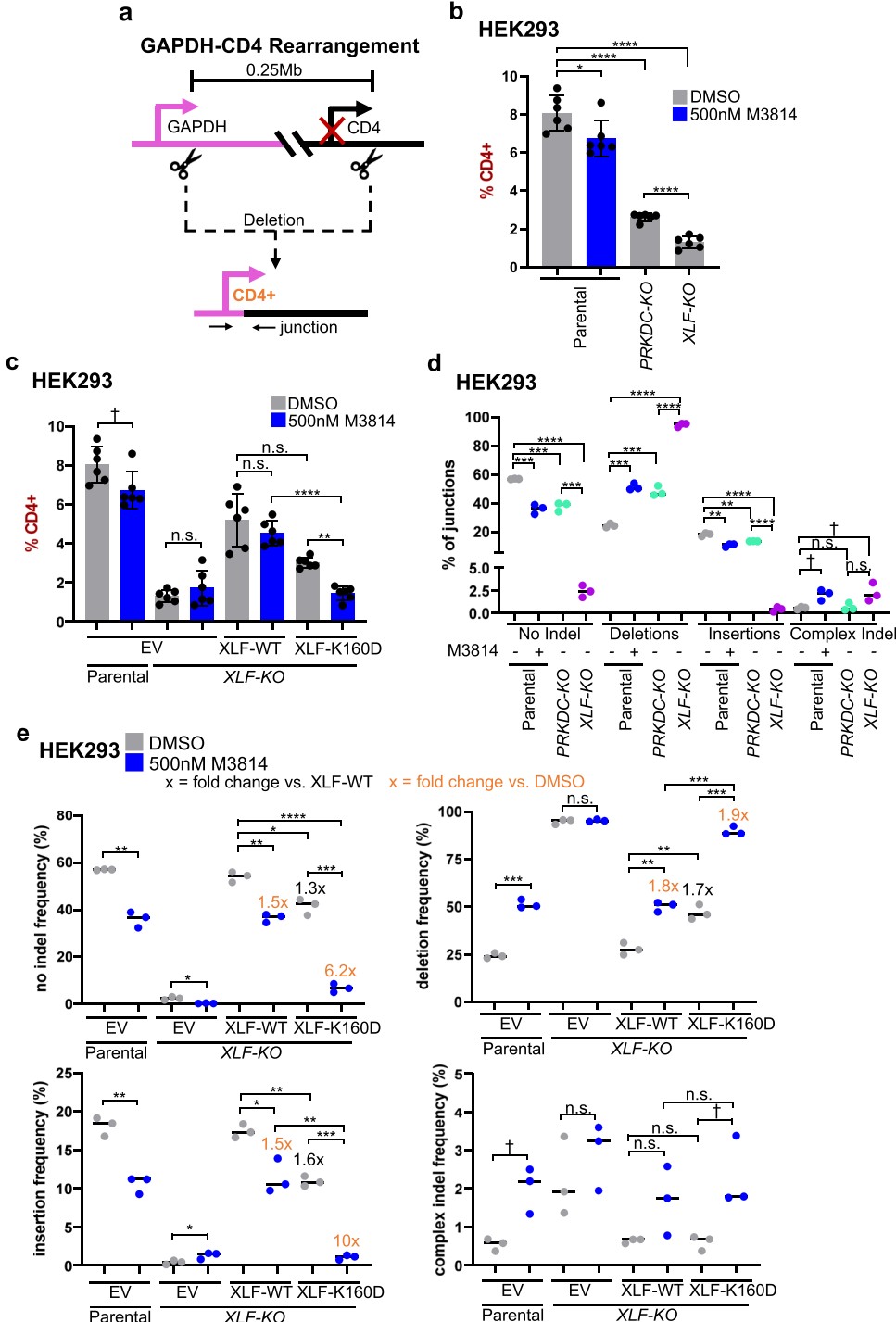

evaluated the nature of such insertions. Beginning with insertion sizes, we found that in the parental, M3814 treated, and *PRKDC-KO* cells the +1 and +2 insertions were predominant, and ≥3 nt insertions were rare (Fig. 4c). For *XLF-KO* cells, with or without M3814 treatment, the ≥3 nt insertions were substantially more prevalent (Fig. 4c, Supplementary Fig. 4a), but we note that the overall frequency of insertions is markedly reduced in these cells (Fig. 3d).

We then considered whether the insertions may result from staggered Cas9 DSBs that cause 5′ overhangs, which are relatively less common than Cas9 blunt DSBs, but nonetheless are readily

detectable[26]. Accordingly, the insertions could be caused by filling-in the 5′ overhangs, and subsequent EJ of the blunt DNA ends. Several studies of insertion sequences with Cas9 DSBs support this model[26,44,45]. Although, it is important to note that since this model is based on inferences from the insertion sequences, other mechanisms are possible, such as non-templated nucleotide addition or 3′ overhang fill-in. For the GAPDH-CD4 rearrangement, staggered Cas9 DSBs at the *GAPDH* site would be expected to cause a 1 nt C-insertion and a 2 nt CG-insertion for 5′ overhangs of 1 and 2 nt, respectively (Supplemental Fig. 4b). To determine the frequency of these events, we examined the

**Fig. 3 Inhibiting DNAPKcs kinase activity affects EJ junction patterns to a greater degree when combined with XLF-K160D. a** Shown is the GAPDH-CD4 rearrangement assay, which uses the endogenous genes (not to scale). HEK293 cells are transfected with expression vectors for Cas9 and the two sgRNAs, with complementing vector(s) or empty vector (EV). CD4 frequencies are normalized to transfection efficiency with parallel GFP transfections. *CD4+* cells are sorted using FACS and used to analyze the GAPDH-CD4 junction patterns by amplicon deep sequencing. **b** Loss of DNAPKcs and XLF leads to decreased *CD4+* cells. HEK293 cells were treated with DMSO or 500 nM M3814. *n* = 6 biologically independent transfections. Statistics with unpaired two-tailed *t* test using Holm–Sidak correction. ****$P$ < 0.0001, *$P$ = 0.0349. **c** Combining 500 nM M3814 and XLF-K160D causes a decrease in *CD4+* cells. *n* = 6 biologically independent transfections. Statistics as in (**b**). ****$P$ < 0.0001, **$P$ = 0.00128, †$P$ = 0.0349 but not significant after correction (unadjusted *P*-value), n.s. not significant. **d** DNAPKcs is less important to promote No Indel EJ and insertions and suppress deletions compared to XLF. Shown are four types of EJ junctions from amplicon deep sequencing. *n* = 3 biologically independent transfections. Statistics as in (**b**). ****$P$ < 0.0001, No Indel Parental DMSO vs. Parental 500 nM M3814 ***$P$ = 0.00083, No Indel Parental DMSO vs. *PRKDC-KO* ***$P$ = 0.00083, No Indel *PRKDC-KO* vs. *XLF-KO* ***$P$ = 0.000141, Deletions Parental DMSO vs. Parental 500 nM M3814 ***$P$ = 0.000113, Deletions Parental +DMSO vs. *PRKDC-KO* ***$P$ = 0.000391, Insertions Parental DMSO vs Parental 500 nM M3814 **$P$ = 0.00273, Insertions Parental DMSO vs. *PRKDC-KO* **$P$ = 0.00273, Complex Indel Parental DMSO vs Parental 500 nM M3814 †$P$ = 0.0149 but not significant after correction (unadjusted *P*-value)., Complex Indel Parental +DMSO vs *XLF-KO* †$P$ = 0.0491 but not significant after correction (unadjusted *P*-value), n.s. not significant. **e** DNAPKcs kinase inhibition affects EJ junctions to a greater degree when combined with XLF-K160D. The frequency of the four types of EJ junctions in (**d**) are shown. *n* = 3 biologically independent transfections. Statistics as in (**b**). No Indel: ****$P$ < 0.0001, ***$P$ = 0.000481, Parental EV DMSO vs 500 nM M3814 **$P$ = 0.00166, *XLF-KO* XLF-WT DMSO vs 500 nM M3814 **$P$ = 0.00166, *$P$ = 0.107. Deletions: Parental EV DMSO vs 500 nM M3814 ***$P$ = 0.000284, *XLF-KO* XLF-WT 500 nM M3814 vs. XLF-K160D 500 nM M3814 ***$P$ = 0.000228, *XLF-KO* XLF-K160D DMSO vs. 500 nM M3814 ***$P$ = 0.000312. Insertions: ***$P$ = 0.000106, Parental EV DMSO vs. 500 nM M3814 **$P$ = 0.00564, *XLF-KO* XLF-WT DMSO vs. XLF-K160D DMSO **$P$ = 0.00245, *XLF-KO* XLF-WT 500 nM M3814 vs. XLF-K160D 500 nM M3814 **$P$ = 0.00516, *$P$ = 0.023. Complex Indel: Parental EV DMSO vs. 500 nM M3814 †$P$ = 0.0149 but not significant after correction (unadjusted *P*-value), *XLF-KO* XLF-K160D DMSO vs. 500 nM M3814 †$P$ = 0.0346 but not significant after correction (unadjusted *P*-value). Data are represented as mean values ± SD.

sequences of the inserted nucleotides, and found the +1 and +2 insertions were almost entirely consistent with such staggered Cas9 DSBs for the parental, M3814 treated, and *PRKDC-KO* cells (Supplementary Fig. 4b, c). For *XLF-KO* cells, with or without M3814, a substantial fraction of the +1 and +2 insertions were also consistent with staggered Cas9 DSBs, but other sequences were also observed (Supplementary Fig. 4b–d). Finally, for the relatively rare ≥3 nt insertions, approximately half of these junctions are also consistent with staggered Cas9 DSBs for the parental cell line, but not the other cells (*PRKDC-KO*, *XLF-KO*, M3814 treated, Supplementary Fig. 4b–d).

The above findings indicate that XLF and DNAPKcs are important for insertions caused by staggered Cas9 DSBs, which we next evaluated with another approach. Specifically, we adapted the EJ7-GFP reporter to examine 1 nt insertions (+1 EJ, EJ7+1-GFP, Fig. 4d). The reporter construct itself is the same as EJ7-GFP, as is the position of the 3′ DSB site, but the 5′ DSB site is targeted 1 nt upstream. Accordingly, GFP would be restored if Cas9 at the 5′ site causes a staggered DSB with a 1 nt 5′ overhang that is filled-in and then repaired by EJ with the distal 3′ blunt DSB end (Fig. 4d, Supplementary Fig. 5a).

Using the EJ7+1-GFP reporter, we examined the influence of DNAPKcs and XLF on +1-EJ events in HEK293 cells. With *PRKDC-KO* and *XLF-KO* cells, we found a significant decrease in such EJ vs. parental HEK293 cells, and that *PRKDC-KO* cells showed a significantly higher frequency vs. *XLF-KO* cells (Fig. 4d). We also interrogated the influence of DNAPKcs in combination with XLF-K160D, and found similar results as with No Indel EJ (Figs. 1, 2), in that combined loss of DNAPKcs and XLF-K160D caused a marked decrease of +1-EJ (Supplementary Fig. 5b), and M3814 treatment in XLF-K160D cells had a substantially greater effect on +1-EJ vs. XLF-WT expressing cells (Supplementary Fig. 5c). Since +1-EJ events are likely caused by blunt EJ after filling-in the 5′ overhang of the staggered Cas9 5′ DSB, and since No Indel EJ is also the product of blunt DSB EJ, we suggest that the XLF-K160D mutant causes a marked requirement for DNAPKcs and its kinase activity for blunt DSB EJ.

**DNAPKcs becomes essential for No Indel EJ in combination with other XLF mutants with disrupted binding interfaces, and depends on the DNAPKcs-ABCDE phosphorylation sites cluster.** Recent cryo-EM structures of LR and SR C-NHEJ

complexes, and other stoichiometric analyses, indicate that a single XLF homodimer can bridge two C-NHEJ complexes bound to DNA ends[8,9,46]. Specifically, the XLF homodimer appears to form this bridge via interactions of each XLF monomer with one molecule each of XRCC4 and KU[8,9,46]. In the above analysis, we used XLF-K160D, which is in the coiled-coil dimerization domain[31]. We speculate that this mutation weakens XLF-mediated bridging of two C-NHEJ complexes, but certainly this mutation could be disrupting XLF function by other mechanisms. Thus, we sought to examine other mutations in XLF. Specifically, we next posited that disrupting the interaction interfaces of XLF with XRCC4 and KU would also weaken XLF-mediated bridging of two C-NHEJ complexes, and hence also reveal a greater requirement for DNAPKcs for No Indel EJ.

To test this hypothesis, we examined several XLF mutants. We examined two mutations that disrupt the leucine-lock interaction with XRCC4: XLF-L115A and XLF-L115D[31,33,46–50]. Both XLF-L115A and L115D fail to pull-down XRCC4 by co-immunoprecipitation, and fail to mediate XLF/XRCC4-mediated DNA tethering[31,33,46–51]. However, XLF-L115A, but not L115D, retains the ability to activate XRCC4/LIG4 ligase activity[51]. These data indicate that XLF-L115A and L115D both cause loss of the interaction with XRCC4, but L115D also disrupts the ability for XLF to activate XRCC4/LIG4 activity[51]. In contrast, in a study using bio-layer inferometry, XLF-L115A, but not L115D, was shown to retain a robust interaction with XRCC4, which supports an alternative model that L115A retains at least a partial interaction with XRCC4[46]. In any case, both the L115A and L115D mutations affect the XRCC4 binding interface. We also tested two mutants that disrupt the KU Binding Motif (ΔKBM that deletes residues 287–299, and XLF-4KA with four conserved lysine residues in the KBM mutated to alanine, which disrupts the interaction with KU)[31,52] (Fig. 5a). Finally, we also examined a combination mutant that disrupts both binding interfaces (L115A/4KA), and another mutant in the homodimer interface (K160A) that is designed to weaken the K160/D161 salt-bridge (Supplementary Fig. 6a, b). An additional benefit of testing these mutants is that they are expressed at a similar level as XLF-WT, although XLF-K160A levels are modestly lower, but nonetheless are higher than XLF-K160D (Fig. 5a, b, Supplementary Fig. 6a, b).

First, we examined the capacity for these various XLF mutants to promote No Indel EJ with and without M3814 treatment (500 nM, as described above), using *XLF-KO* HEK293 cells.

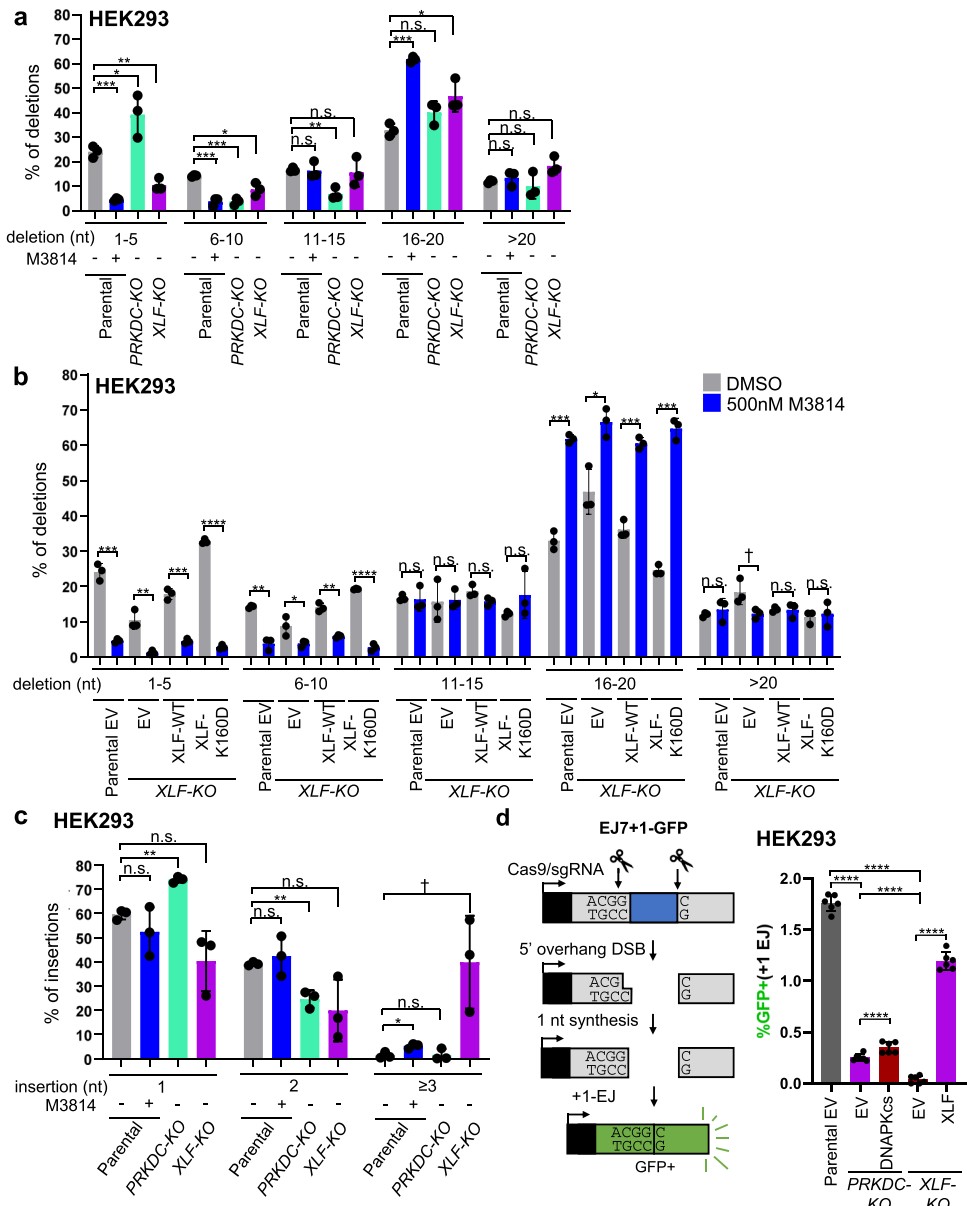

**Fig. 4 The influence of DNAPKcs and XLF on the sizes of deletions and insertions at EJ junctions, and on a +1 insertion EJ event likely caused by staggered Cas9 DSBs. a** Shown is the frequency of deletion sizes for parental cells with or without M3814, *PRKDC-KO*, and *XLF-KO* cells for the deletion events shown in Fig. 3d. *n* = 3 biologically independent transfections. Statistics with unpaired two-tailed *t* test using Holm–Sidak correction. 1–5 nt deletion: ***P = 0.000475, **P = 0.00506, *P = 0.0447. 6–10nt deletion: Parental DMSO vs 500 nM M3814 ***P = 0.000937, Parental DMSO vs. *PRKDC-KO* ***P = 0.000596, *P = 0.0247. 11–15 nt deletion: **P = 0.00824. 16–20 nt deletion: ***P = 0.000189, *P = 0.0491. n.s.= not significant. **b** Shown is the frequency of deletion sizes for parental and *XLF-KO* cells with or without M3814 for the deletion events shown in Fig. 3e. *n* = 3 biologically independent transfections. Statistics as in (**a**). 1–5nt deletion: ****P < 0.0001, Parental EV DMSO vs. 500 nM M3814 ***P = 0.000473, *XLF-KO* XLF-WT DMSO vs. 500 nM M3814 ***P = 0.000473, **P = 0.00374. 6–10nt deletion: ****P < 0.0001, *Parental* EV DMSO vs. 500 nM M3814 **P = 0.00127, *XLF-KO* XLF-WT DMSO vs. 500 nM M3814 **P = 0.00127, *P = 0.041. 11–15nt deletion: n.s.= not significant. 16–20 nt deletion: Parental EV DMSO vs. 500 nM M3814 ***P = 0.000189, *XLF-KO* XLF-WT DMSO vs. 500 nM M3814 ***P = 0.000274, *XLF-KO* XLF-K160D DMSO vs. 500 nM M3814 ***P = 0.000121, *P = 0.0104. >20 nt deletion: †P = 0.0478 but not significant after correction (unadjusted *P*-value), n.s.= not significant. **c** Shown is the frequency of insertion sizes for parental cells with or without M3814, *PRKDC-KO*, and *XLF-KO* cells for the insertion events shown in Fig. 3d. *n* = 3 biologically independent transfections. Statistics as in (**a**). 1 nt insertion: **P = 0.00147. 2 nt insertion: **P = 0.00692. ≥3 nt insertion: *P = 0.0446, †P = 0.0257 but not significant after correction (unadjusted *P*-value). n.s.= not significant. **d** DNAPKcs is less important to promote EJ with 1 nt insertions (+1 EJ) compared to XLF. Shown is the EJ7+1-GFP reporter to measure +1 EJ, which is the same chromosomal reporter as in Fig. 1, but using a different 5′ sgRNA. *n* = 6 biologically independent transfections. Statistics as in (**a**). ****P < 0.0001. Data are represented as mean values ± SD.

Without M3814 treatment, we found that XLF-L115A, XLF-4KA, and XLF-ΔKBM were each able to promote No Indel EJ to the same degree as XLF-WT, whereas XLF-K160A and XLF-L115A/ 4KA showed a modest reduction, and XLF-L115D showed a marked reduction (Fig. 5a, Supplementary Fig. 6a). Next, when we examined each of these XLF mutants in combination with DNAPKcs kinase inhibition (M3814 treatment), we found that all of the XLF mutants caused a significant decrease in No Indel EJ compared to XLF-WT (Fig. 5a, Supplementary Fig. 6a). Furthermore, M3814 treatment caused a greater fold decrease in No

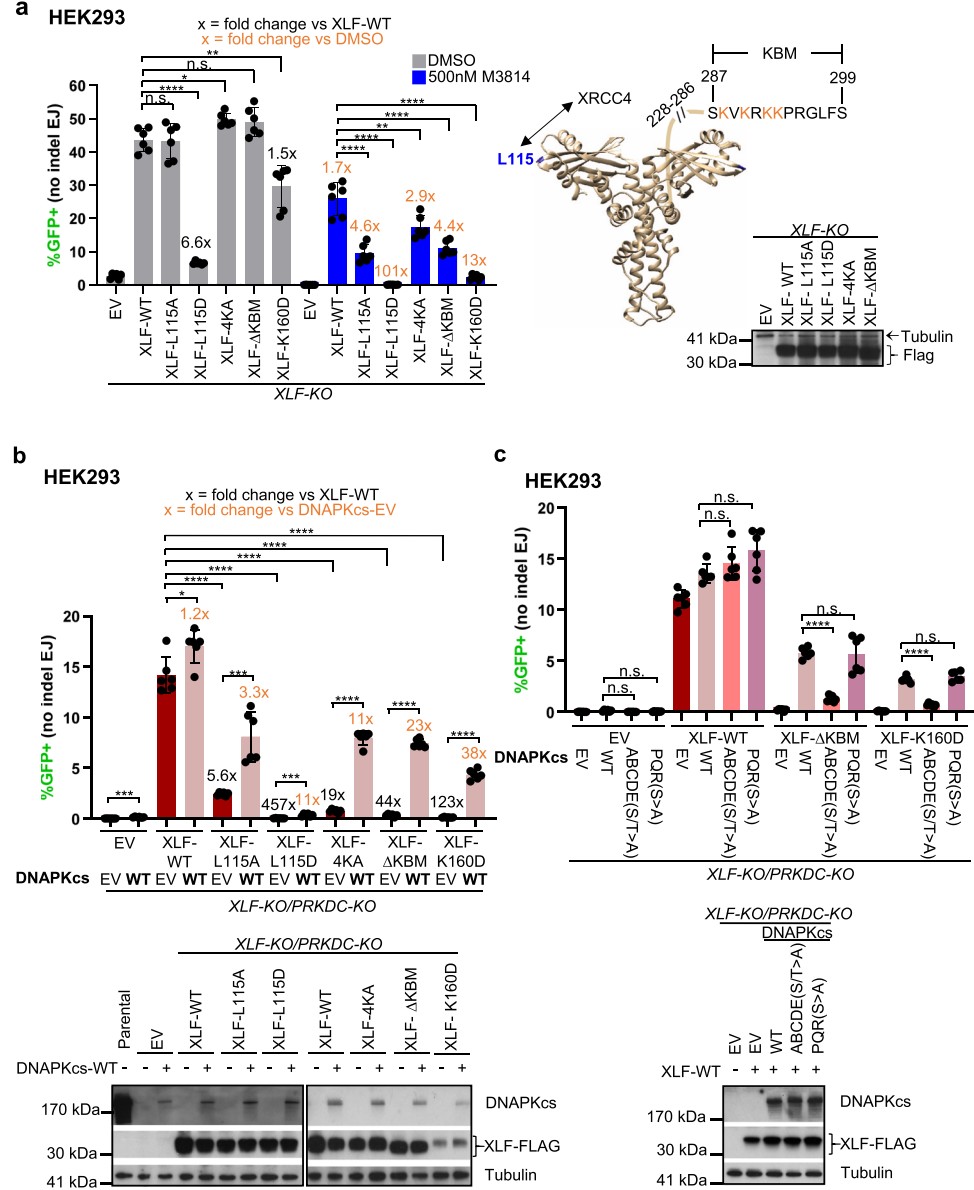

**Fig. 5 DNAPKcs becomes essential for No Indel EJ in combination with other mutants in XLF binding interfaces, and depends on the DNAPKcs-ABCDE phosphorylation sites cluster. a** DNAPKcs kinase inhibition has a markedly greater effect on No Indel EJ in cells with XLF mutants disrupting the XRCC4 or KU interaction interfaces. $n = 6$ biologically independent transfections. Statistics with unpaired two-tailed $t$ test using Holm-Sidak correction. ****$P < 0.0001$, XLF-WT DMSO vs. XLF-K160D DMSO **$P = 0.00327$, XLF-WT 500 nM M3814 vs. XLF-4KA 500 nM M3814 **$P = 0.00565$, *$P = 0.0131$, n.s. not significant. Also shown is the XLF homodimer (aa 1–227, Protein Data Bank 2R9A, with the C-terminus for one monomer drawn as a line), with the L115 residue important for XRCC4 interaction, Ku Binding Motif (KBM), and 4 lysine (K) residues in the KBM highlighted. Immunoblot shows levels of FLAG-tagged XLF-WT and mutants. **b** DNAPKcs is required to promote No Indel EJ in cells with XLF mutants disrupting the XRCC4 and KU interaction interfaces. $n = 6$ biologically independent transfections. Statistics as in (**a**). ****$P < 0.0001$, XLF-EV DNAPKcs-EV vs DNAPKcs-WT ***$P = 0.000169$, XLF-L115A DNAPKcs-EV vs DNAPKcs-WT ***$P = 0.000666$, XLF-L115D DNAPKcs-EV vs DNAPKcs-WT ***$P = 0.000697$, *$P = 0.0162$ Immunoblots show levels of FLAG-tagged XLF-WT and mutants, and DNAPKcs. **c** DNAPKcs-ABCDE phosphorylation cluster is required to promote No Indel EJ in cells expressing XLF-ΔKBM and XLF-K160D. $n = 6$ biologically independent transfections. Statistics as in (**a**). ****$P < 0.0001$, n.s. not significant. Immunoblot shows levels of DNAPKcs-WT, DNAPKcs-ABCDE(S/T>A), and DNAPKcs-PQR(S>A). Data are represented as mean values ± SD.

Indel EJ with each of these mutants, as compared to XLF-WT (Fig. 5a, Supplementary Fig. 6a).

We also examined the XLF mutants in the presence or absence of DNAPKcs using *XLF-KO/PRKDC-KO* cells. We found that in the absence of DNAPKcs, all the XLF mutants caused a significant decrease in No Indel EJ compared to XLF-WT, and transient expression of DNAPKcs with all the XLF mutants caused a significant increase in No Indel EJ (Fig. 5b, Supplementary Fig. 6b). Furthermore, loss of DNAPKcs caused a greater fold

decrease with each of the XLF mutants, as compared to XLF-WT. For example, whereas in the *XLF-KO* cell line the XLF-ΔKBM mutant showed no defect in No Indel EJ, in the *XLF-KO/PRKDC-KO* cells this mutant was reduced 44-fold compared to XLF-WT, and transient expression of DNAPKcs was able to restore such EJ 23-fold (Fig. 5b). As another example, the XLF-L115A/4KA mutant caused a 1.8-fold decrease in the *XLF-KO* cell line, but a 241-fold decrease in the *XLF-KO/PRKDC-KO* cell line, with DNAPKcs expression causing a 65-fold increase (Supplementary

Fig. 6b). These findings indicate that a series of XLF mutants that disrupt its interactions with XRCC4 and KU, as well as an additional mutant in the homodimer interface (K160A), each cause an enhanced requirement for DNAPKcs and its kinase activity for No Indel EJ. Finally, we also tested two previously described XLF mutants (6S/T>A, 5S/T>D) in phosphorylation sites in the C-terminus[49,53], which we found had no obvious phenotype (Supplementary Fig. 6a, b).

Given that a main phosphorylation target of DNAPKcs appears to be itself at several sites, including the ABCDE and PQR clusters[36,37,54], we posited that such autophosphorylation site clusters may be important to promote No Indel EJ in the context of a weakened XLF. To test this hypothesis, we examined No Indel EJ in XLF-KO/PRKDC-KO HEK293 cells expressing XLF-ΔKBM and XLF-K160D in combination with two DNAPKcs mutants: the DNAPKcs-ABCDE(S/T>A) mutant that contains serine/threonine to alanine mutations at six phosphorylation sites (T2609, S2612, T2620, S2624, T2638, and T2647), and DNAPKcs-PQR(S>A) mutant that contains such mutations at five serine residues (S2023, S2029, S2041, S2053, and S2056)[54,55]. Beginning with XLF-KO/PRKDC-KO HEK293 cells with or without XLF-WT, we found that similar to expressing DNAPKcs-WT, the mutants did not substantially affect No Indel EJ (Fig. 5b, c). For XLF-KO/PRKDC-KO HEK293 cells expressing XLF-K160D and XLF-ΔKBM, expression of DNAPKcs-WT markedly promotes No Indel EJ (Fig. 5b), which enabled a functional examination of the DNAPKcs mutants. From this analysis we found that, when combined with either the XLF-K160D or XLF-ΔKBM, DNAPKcs-PQR(S>A) was not different from DNAPKcs-WT, whereas the DNAPKcs-ABCDE(S/T>A) mutant showed a marked defect in No Indel EJ (Fig. 5c). This finding indicates that the ABCDE phosphorylation sites cluster of DNAPKcs is critical for No Indel EJ when combined with a weakened XLF.

Finally, a recent cryo-EM study identified a possible DNAPKcs dimerization interface that involves the residues 898–900, and mutation of these residues to alanine (898–900>A) was shown to cause a defect in V(D)J recombination[9]. Thus, we examined the 898–900>A mutant in a similar experiment as the phosphorylation sites mutants described above. From this analysis, we found that DNAPKcs-898–900>A showed a marked defect in promoting No Indel EJ in XLF-KO/PRKDC-KO HEK293 cells expressing XLF-K160D and XLF-ΔKBM (Supplementary Fig. 6c). This finding indicates that a DNAPKcs dimerization interface is important for No Indel EJ. However, of course the 898–900>A mutations could be having other effects apart from dimerization per se.

**DNAPKcs kinase inhibition causes an increase in HDR and structural variants (SVs), and IR sensitivity that is not further enhanced by XLF loss.** Thus far our study has focused on the influence of the C-NHEJ factors DNAPKcs and XLF on EJ of blunt DSBs, but C-NHEJ also has other roles in DSB repair, such as suppression of homology-directed repair (HDR). Namely, loss of C-NHEJ factors (e.g., KU, DNAPKcs, and XRCC4), and inhibition of DNAPKcs kinase activity, have been shown to cause an increase in HDR[22,54,56,57]. Thus, we sought to compare our above findings of the influence of DNAPKcs and XLF on blunt DSB EJ vs. suppression of HDR. To examine HDR, we used the LMNA-HDR assay, which measures HDR at the endogenous LMNA gene[58,59] (Fig. 6a). This reporter involves co-transfecting an sgRNA/Cas9 expression vector to target a Cas9 DSB in LMNA exon 1, along with a plasmid donor that contains the LMNA exon 1 fused in frame with the fluorescent protein mRuby2[58,59]. Thus, HDR using the plasmid donor causes mRuby2+ cells that can be

detected via flow cytometry (Fig. 6a)[58,59]. This assay has been validated to be dependent on the key HDR factor PALB2[58,59]. We performed an independent validation using siRNA depletion of the HDR factor BRCA2 (siBRCA2) and found a marked reduction in mRuby2+ cells, compared to non-targeting siRNA (siCtrl, 12-fold, Fig. 6b). As a comparison, we found that No Indel EJ was only modestly affected by siBRCA2 treatment (1.2-fold, Fig. 6b).

Using this assay, we examined HDR frequencies in the PRKDC-KO, XLF-KO, and XRCC4-KO HEK293 cells, as well as parental cells treated with M3814. With each of these three KO cell lines, we found a significant increase in HDR, compared to the parental HEK293 cells (Fig. 6c). In comparing the mutant lines, the PRKDC-KO cells showed significantly lower levels of HDR vs. XLF-KO, and XRCC4-KO, however the relative differences were modest (Fig. 6c). We then examined the effect of M3814 treatment in the parental cells, which we found significantly increased HDR with 250 nM, with no further increase with higher concentrations of M3814 (500 nM and 1000 nM, Fig. 6d). This result indicates that the effect of M3814 on increasing HDR is saturated at 250 nM, which is notably different from the relatively mild effects of this concentration No Indel EJ, compared to 500 nM and 1000 nM (Fig. 2a). Furthermore, the fold increase of HDR caused by 500 nM M3814 in parental cells was similar to the effect of XLF loss, whereas No Indel EJ was only modestly affected by 500 nM M3814 vs. the marked defect in XLF-KO cells (Fig. 2a). Accordingly, we observe distinctions in the relative effect of M3814 on No Indel EJ vs. suppression of HDR. As a control, we tested overexpression of XLF in parental cells with and without M3814 treatment, and found a slight increase in HDR with XLF overexpression with 250 nM M3814, but no effects without M3814 or the higher concentrations (Supplementary Fig. 7a).

We then sought to examine the effects on HDR of combined disruption of DNAPKcs and XLF, as well as with the XLF-K160D mutant. Interestingly, we found that the XLF-KO/PRKDC-KO cells did not show an increase in HDR frequencies compared to the parental cells, and transient expression of XLF and DNAPKcs had no effect on HDR in XLF-KO/PRKDC-KO cells (Supplementary Fig. S7a), which we speculate may be due to an adaptation response in these cells to downregulate HDR. In any case, since the XLF and DNAPKcs expression vectors had no effect on HDR these cells, we were unable to examine mutants of these factors in this context. However, since XLF-WT expression suppressed HDR in the XLF-KO cells (Fig. 6c), we were able to examine XLF mutants alone, and in combination with M3814 treatment. From this analysis, we found that M3814 treatment in the XLF-KO cells did not cause an increase in HDR (Fig. 6d). From comparison of XLF-WT vs. XLF-K160D in XLF-KO cells, XLF-WT was significantly more proficient than XLF-K160D in suppressing HDR both in the absence of M3814 as well as at the lowest concentration (250 nM), but with higher concentrations of M3814 (500 nM and 1000 nM), XLF-WT and XLF-K160D were not statistically different (Fig. 6d). These findings indicate that DNAPKcs kinase inhibition causes an increase in HDR that is not further enhanced by XLF loss.

Another reported outcome of sgRNA/Cas9 DSBs is large deletion mutations[60,61], such that we sought to examine the influence of XLF and DNAPKcs kinase inhibition on such events. To study such deletions in human cells, we developed an assay using GFP fused to a protein degradation sequence (GFPd2)[62], which shortens its half-life, thereby enabling its use as a mutagenesis reporter[63]. We integrated GFPd2 into the FRT site of HEK293 cells, using the Flp recombinase, generated Cas9 DSBs using the PiggyBac system for stable expression of sgRNAs/Cas9[60], and monitored the frequency of GFP-negative (GFP-neg cells). We induced DSBs at this locus in three positions: DSB-H

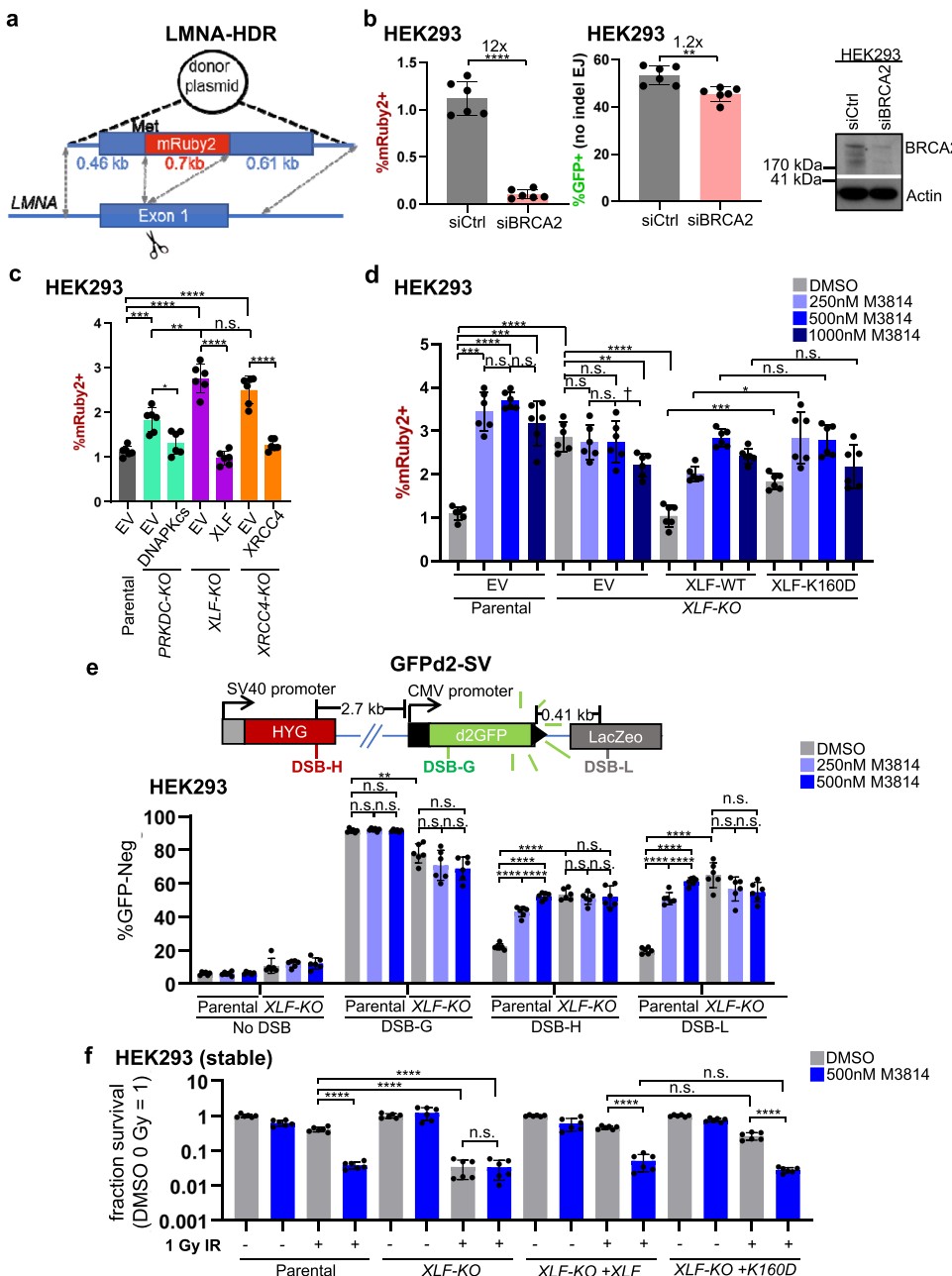

**Fig. 6 DNAPKcs kinase inhibition causes an increase in HDR and structural variants (SVs), and IR sensitivity that is not further enhanced by XLF loss.**
**a** Shown is the LMNA-HDR reporter (not to scale) involving a Cas9/sgRNA that induces a DSB in *LMNA* exon 1, and a plasmid donor such that HDR causes mRuby2 to be expressed from the *LMNA* locus. The mRuby2 frequencies are normalized to transfection efficiency with parallel GFP transfections. **b** BRCA2 is required for HDR by the LMNA-HDR assay, but has modest effects on No Indel EJ. Cells were transfected with siRNA that is non-targeting (siCtrl) or targets BRCA2 (siBRCA2). $n = 6$ biologically independent transfections. Statistics with unpaired two-tailed $t$ test using Holm–Sidak correction.
****$P < 0.0001$, **$P = 0.00305$. Immunoblot shows levels of BRCA2. **c** DNAPKcs, XLF, and XRCC4 suppress HDR to similar degrees. $n = 6$ biologically independent transfections. Statistics as in (**b**). ****$P < 0.0001$, ***$P = 0.00069$, **$P = 0.00117$, n.s. not significant. **d** Suppression of HDR. $n = 6$ biologically independent transfections. Statistics as in (**b**). ****$P < 0.0001$, Parental EV DMSO vs. 1000 nM M3814 ***$P = 0.000493$, *XLF-KO* XLF-WT DMSO vs. XLF-K160D DMSO ***$P = 0.000404$, *XLF-KO* EV DMSO vs. 1000 nM M3814 **$P = 0.0203$, *XLF-KO* XLF-WT 250 nM M3814 vs XLF-K160D 250 nM M3814 *$P = 0.0249$, †$P = 0.0386$ but not significant after correction (unadjusted $P$-value), n.s. not significant. **e** Suppression of SVs. Shown is the GFPd2-SV reporter (not to scale) that is chromosomally integrated to an FRT site in HEK293 cells. Stable expression of Cas9 and various sgRNAs (DSB-H, DSB-G, and DSB-L) can induce GFP-negative cells (GFP-neg). $n = 6$. Statistics as in (**b**). ****$P < 0.0001$, **$P = 0.00136$, n.s. not significant. **f** IR sensitivity.
HEK293 stable cell lines were treated with DMSO or M3814, and 0 Gy or 1 Gy IR dose, and plated to form colonies. Fraction clonogenic survival was determined relative to DMSO 0 Gy for each cell line. $n = 6$ biologically independent wells of seeded cells. Statistics as in (**b**). ****$P < 0.0001$, n.s.= not significant. Error bars = SD. Data are represented as mean values ± SD.

that is 2.7 kb upstream from the start of the GFP cassette, DSB-L that is 0.41 kb downstream from the end of the GFP cassette, and DSB-G that is within the GFP coding sequence (Fig. 6e). Each of these sgRNAs were confirmed to induce indels at the predicted target site (Supplementary Fig. 7b). As compared to an sgRNA control that does not target the locus (No DSB), we found that each of these DSB sites induced GFP-neg cells, with DSB-G causing the highest frequency of GFP-neg cells, compared to DSB-H and DSB-L (Fig. 6e). These findings indicate that DSBs far from the GFP expression cassette can induce GFP-neg cells, which we then posited would be associated with the loss of a substantial segment of the reporter locus. To test this hypothesis, we examined three sites in the reporter by quantitative PCR (qPCR): one downstream of DSB-H (site A), and two upstream from DSB-L (sites B and C) (Supplementary Fig. 7b). We found that the GFP-neg cells induced by either DSB-H or DSB-L showed a significant decrease in amplicons from all three sites, compared to those induced by DSB-G that is within the GFP coding sequence (Supplementary Fig. 7b). Thus, DSB-H and DSB-L induce GFP-neg events that are associated with the loss of a substantial segment of the reporter locus. The most likely explanation for such events is a large deletion, as described[60,61], however since more complex events could conceivably cause such genetic loss, we refer to them as structural variants (SVs), based on the definition of a genetic alteration involving >50 bp[64]. As such, we named this reporter GFPd2-SV.

Using this assay, we then examined the influence of XLF loss and M3814 treatment on the frequency of GFP-neg cells induced by DSBs at all three sites. Beginning with DSB-G that is within the GFP coding sequence, M3814 treatment had no effect on the induction of GFP-neg cells, and XLF loss caused a modest decrease in the frequency of these events (Fig. 6e). In contrast, for the DSBs induced away from the GFP expression cassette (DSB-H and DSB-L), both M3814 treatment and XLF loss caused a significant increase in GFP-neg cells (Fig. 6e), indicating that DNAPKcs kinase activity and XLF are important to suppress DSB-induced SVs. We then compared the fold increase in SVs, finding that 500 nM M3814 treatment had a similar effect as XLF loss, and 250 nM M3814 was only modestly lower than either (Fig. 6e, both DSB-H and DSB-L). Furthermore, M3814 treatment had no effect on such SVs in the *XLF-KO* cells (Fig. 6e). Accordingly, similar to our findings with HDR, suppression of DSB-induced SVs is sensitive to inhibition of DNAPKcs kinase activity in a manner that is not further enhanced by loss of XLF.

Finally, we examined effects of XLF loss, XLF-K160D expression, and M3814 treatment on clonogenic survival following ionizing radiation (IR). We used colony-forming assays for this analysis, which is a common approach for examining reproductive cell death[65]. Although, we note that a limitation of colony-forming assays is that they do not provide information on the rate of cell proliferation. For these experiments, we first created *XLF-KO* HEK293 cell lines with stable expression of XLF-WT and XLF-K160D (Supplementary Fig. 7c, control cell lines stably transfected with EV). As with the transient expression experiments, stable expression of XLF-K160D combined with M3814 treatment causes a marked reduction in No Indel EJ, compared to XLF-WT (Supplementary Fig. 7c). Using these cell lines, we first examined fraction clonogenic survival using two doses of IR (0.5 and 1 Gy), both with and without M3814 (500 nM) (Fig. 6f, Supplementary Fig. 8a). We found that the addition of M3814 to *XLF-KO* cells did not further enhance IR sensitivity with either IR dose (Fig. 6f, Supplementary Fig. 8a). Next, we examined IR treatment in *XLF-KO* with stable expression of XLF-K160D and found a modest but significant decrease compared to stable expression of XLF-WT for 1 Gy IR,

but no significant difference for 0.5 Gy IR (Fig. 6f). Lastly, we examined combinations of M3814 treatment with both the XLF-WT and XLF-K160D cell lines, and found no significant difference in clonogenic survival after either IR dose (Fig. 6f, Supplementary Fig 8a). These results are in accordance with the similar increase in HDR in M3814 treated XLF-WT and XLF-K160D (Fig. 6d), but are distinct from the findings that combining XLF-K160D with M3814 causes a marked loss of No Indel EJ (Fig. 2a). Altogether, these findings indicate that suppression of HDR, SVs, and IR-resistance at both 0.5 and 1 Gy are each sensitive to kinase inhibition of DNAPKcs in a manner that is not further enhanced by XLF loss.

Since DNAPKcs kinase inhibition (M3814 treatment) failed to cause IR sensitivity with *XLF-KO* cells (Fig. 6f), we then posited that combining genetic loss of DNAPKcs with XLF may show similar results. Thus, we examined clonogenic survival using two doses of IR (0.5 and 1 Gy) with a series of HEK293 cell lines: parental, *PRKDC-KO*, *XLF-KO*, *XRCC4-KO*, and *XLF-KO/PRKDC-KO*. We found that for 0.5 Gy each of these cell lines showed greater sensitivity vs. the parental line, but were not statistically different from each other (Supplementary Fig. 8b). We found similar results using 1 Gy, except that the *XRCC4-KO* showed reduced clonogenic survival vs. the other cell lines (Supplementary Fig 8c). Thus, similar to the results with M3814 treatment, genetic loss of DNAPKcs failed to cause IR sensitivity in cells without XLF.

## Discussion

We sought to define the role of DNAPKcs on chromosomal DSB repair, with a focus on EJ of blunt DSBs, because other components of C-NHEJ (e.g., XLF and XRCC4) have been shown to be critical for such EJ[31]. Furthermore, we have examined EJ of blunt DSBs to model repair of ligatable ends that may not be stabilized by an annealing intermediate[66]. As well, examining the repair of Cas9 DSBs is inherently significant to develop approaches to gene editing[1]. To assess blunt DSB EJ, we examined two types of repair outcome: No Indel EJ between distal ends of two Cas9 blunt DSBs, as well as insertion mutations that are consistent with staggered Cas9 DSBs causing 5′ overhangs that are filled-in to generate a blunt end. We found similar results with these two types of repair outcomes. Namely, DNAPKcs promotes these EJ events, but is less important than XLF, and its role is substantially magnified in the presence of a weakened XLF. Considering our findings in the context of recent cryo-EM structures[8,9], we suggest that DNAPKcs is important to position a weakened XLF into an LR complex that can efficiently transition into an SR complex to mediate blunt DSB EJ, but that a weakened XLF cannot support a functional SR complex de novo (Fig. 7). In contrast, with XLF-WT, the DNAPKcs LR complex does not appear to be absolutely required for establishing a functional SR complex for blunt DSB EJ. We speculate that the SR complex assembles de novo via a series of relatively weak protein-protein interactions, such that disrupting any of these interactions would abolish SR complex assembly unless first stabilized via DNAPKcs within the LR complex.

Based on these findings, the requirement for DNAPKcs during chromosomal EJ appears dependent on the specific circumstance. In *Xenopus* extracts, DNAPKcs is required for both establishing LR synapsis and transition to the SR synapsis state[7]. This role of DNAPKcs to establish an LR complex may involve direct dimerization across DSBs. Consistent with this notion, we found that DNAPKcs residues in a proposed dimer interface (898–900)[9] are important for blunt DSB EJ. However, in the context of a chromosome in the cell nucleus, the stability of DSB end synapsis could be facilitated by higher-order chromosome structure[67,68]

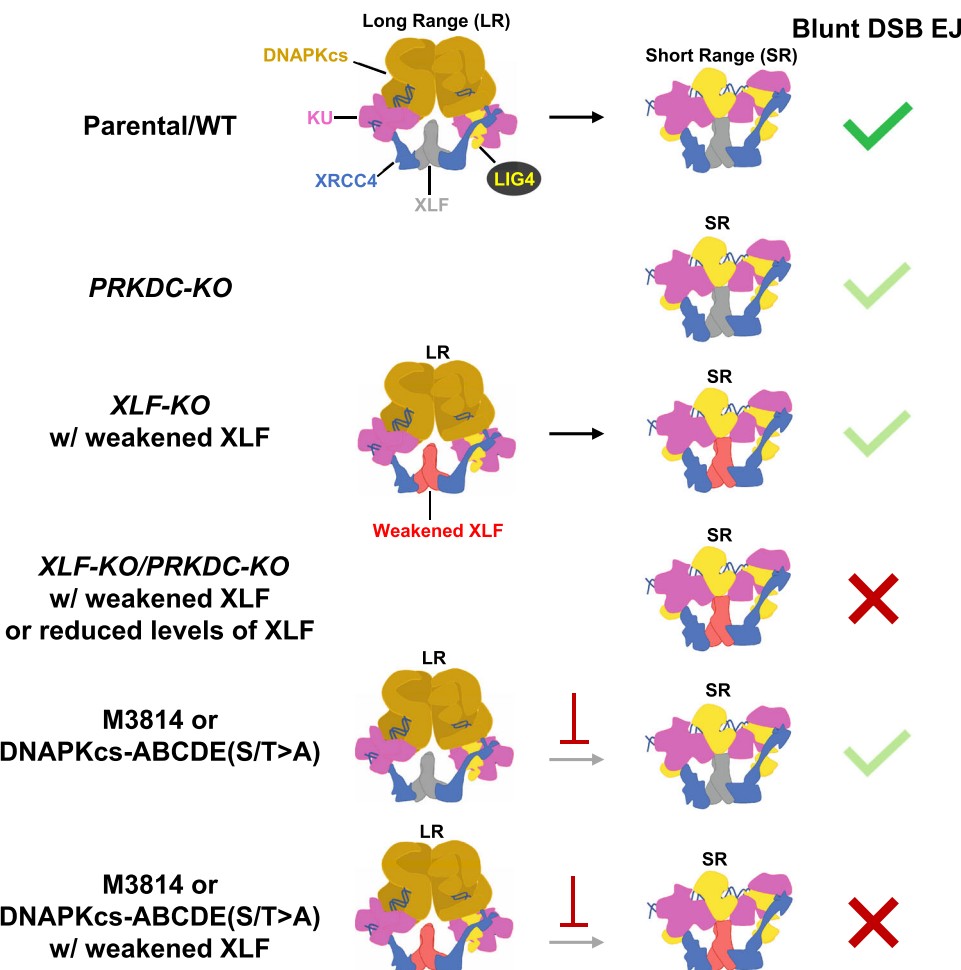

**Fig. 7 Summary.** Model for the DNAPKcs LR C-NHEJ complex being important to stabilize a weakened XLF to transition into a functional SR complex to facilitate blunt DSB EJ. Illustrations adapted from published cryo-EM structures described in the text. Light green checkmark depicts a partial defect, whereas a red X denotes a marked loss of blunt DSB EJ.

that could provide some partial redundancy with the LR complex mediated by DNAPKcs. Namely, in the context of chromatin, the SR complex may be proficient to form de novo and mediate blunt DSB EJ. Similarly, the DNA damage response factor 53BP1 could facilitate long-range end synapsis. Namely, 53BP1 has been proposed to mediate chromosomal end synapsis during class switch recombination and fusions of deprotected telomeres[69], and shows a synthetic defect in V(D)J recombination when combined with XLF loss[70,71]. In summary, 53BP1 and/or chromatin structure may facilitate backup mechanisms for long-range end synapsis during C-NHEJ. Such functional redundancy for end synapsis may have evolved to ensure efficient blunt chromosomal DSB EJ[66,68].

We used several approaches to weaken XLF, including six different sets of XLF mutations that each caused a magnified requirement for DNAPKcs and its kinase activity for blunt DSB EJ. The fold-effects of the mutants showed variations, with the XLF-L115D mutation in the XRCC4 interaction interface causing the greatest fold-defect both with and without DNAPKcs. This finding is consistent with the defect for this mutant to activate XRCC4/LIG4 ligase activity[51], which is central to C-NHEJ. Similarly, loss of XRCC4 caused a greater fold-defect in blunt DSB EJ vs. XLF, likely due to the key role of this protein for LIG4 activity[72]. The K160D and L115A/4KA mutants showed modest defects in the presence of DNAPKcs, but showed marked reductions in No Indel EJ when combined with DNAPKcs loss or

with M3814 treatment. The individual mutants L115A, 4KA, ΔKBM, were less severe than L115A/4KA likely due to each mutant retaining binding interfaces with the C-NHEJ complex that are lost with the double L115A/4KA mutant. Similarly, K160A was less severe than K160D, likely due to causing loss of the K160-D161 salt bridge without also causing the charge repulsion of K160D. Finally, we also tested depletion of XLF by shRNA, finding that reduced XLF levels only caused a defect in No Indel EJ in the absence of DNAPKcs, indicating that such EJ is not particularly sensitive to XLF levels, unless DNAPKcs is absent. These findings indicate that the ability for the XLF homodimer to interact with multiple components of the C-NHEJ complex is critical for blunt DSB EJ, particularly when DNAPKcs is disrupted.

In addition to studies with DNAPKcs loss, we also found that DNAPKcs kinase inhibition, and the DNAPKcs-ABCDE(S/T>A) mutant, each show a marked reduction of blunt DSB EJ when combined with a weakened XLF. For example, from amplicon sequencing of EJ junctions, DNAPKcs kinase inhibition combined with XLF-K160D causes a marked reduction in blunt DSB EJ (No Indel EJ and insertions), and a converse increase in deletions. These findings are notable, because several lines of evidence indicate that DNAPKcs kinase inhibition and the DNAPKcs-ABCDE(S/T>A) mutant block DNAPKcs displacement from the LR complex. For one, the LR to SR transition is blocked by DNAPKcs kinase inhibitors in *Xenopus* extracts[7].

DNAPKcs kinase inhibitors have also been shown to cause a stable DNAPKcs complex on DSB ends that are prone to cleavage via the MRE11 nuclease[35]. As well, kinase-dead DNAPKcs shows persistent recruitment to laser DNA damage[37]. Finally, kinase-dead DNAPKcs causes severe defects in mouse embryonic development that is rescued with loss of KU[73], which is also consistent with DNAPKcs kinase inhibition causing its hyper-stabilization on DNA ends. The likely mechanism of such stabilization is via blocking DNAPKcs trans-autophosphorylation. Specifically, mutation of such phosphorylation sites, particularly within the ABCDE cluster, causes persistent recruitment to laser DNA damage[37], hyper-stable complexes on DNA ends with purified proteins[74,75], and persistent association with damaged chromatin[76]. Altogether, we suggest that displacement of DNAPKcs is required for the LR complex to transition into a functional SR complex for blunt DSB EJ, which is particularly essential when combined with a weakened XLF that cannot support a functional SR complex de novo (Fig. 7).

We also examined other roles of DNAPKcs and XLF on DSB repair apart from blunt DSB EJ. For one, we examined deletion sizes at EJ junctions, finding that M3814 treatment (500 nM) caused a substantial shift from short deletions (1–5, 6–10 nt.) to relatively larger deletions (16–20 nt.), which was similar to the effect of XLF loss. Interestingly, M3814 treatment also caused this shift to relatively larger deletions in *XLF-KO* cells. These findings indicate that DNAPKcs kinase activity is important to promote relatively shorter deletions in a manner independent of XLF. As one possible mechanism, DNAPKcs kinase activity could be critical for regulation of end processing nucleases during EJ, such as the Artemis nuclease, to favor short deletions that could be mediated either by C-NHEJ or XLF-independent Alternative EJ. Consistent with this model, DNAPKcs autophosphorylation has been shown to regulate the recruitment of the Artemis nuclease to DNA ends[77].

Additionally, we examined the regulation of HDR, which is known to be suppressed by C-NHEJ[22,54,56,57]. We found that suppression of HDR appears more sensitive to DNAPKcs kinase inhibition, compared to its effects on blunt DSB EJ. For example, a low dose of M3814 (250 nM) was saturating for causing an increase in HDR (vs. 500 and 1000 nM), whereas these higher doses caused a concentration-dependent decrease in blunt DSB EJ. Furthermore, combining 500 nM M3814 with XLF loss causes a similar increase in HDR, compared to the M3814 treatment alone. In contrast, this combination of 500 nM M3814 and XLF loss shows markedly lower blunt DSB EJ, compared to such M3814 treatment alone. We suggest a model whereby DNAPKcs kinase inhibition slows the transition from the LR to the SR complex, but such a delay may not have a marked effect on blunt DSB EJ, at least when combined with XLF-WT that could form a functional SR complex de novo. In contrast, such a delay could be sufficient to stimulate HDR, which likely involves the initiation of DSB end resection that is mediated by the MRE11-RAD50-NBS1 complex with CtIP[35]. Consistent with this notion, DNAPKcs kinase inhibition has been shown to promote MRE11-dependent cleavage of a DNAPKcs complex at DSB ends[35]. Accordingly, efficient/rapid transition of the LR complex to the SR complex may be critical for the suppression of such end resection initiation.

One possible consequence of the deregulation of end resection initiation is genome instability. Namely, resected ends with long ssDNA are likely not readily repaired by C-NHEJ, nor HDR when the sister chromatid is unavailable. Thus, repair of such resected ends may be prone to more mutagenic processes. This model is consistent with our findings of DSB-induced SVs. Namely, similar to the findings with HDR, combining 500 nM M3814 treatment with XLF loss caused a similar level of DSB-induced SVs, as compared to the M3814 treatment alone. Strikingly, we observed the same pattern with clonogenic survival following IR treatment. Accordingly, we speculate that DSB-induced SVs, along with other mutagenic events, could be a major contributing factor to the radiation sensitization caused by DNAPKcs kinase inhibition, rather than being due solely to a reduction in blunt DSB EJ. In conclusion, the role of DNAPKcs for blunt DSB EJ is substantially magnified when XLF is weakened, but DNAPKcs also has distinct functions in regulating other aspects of genome maintenance.

## Methods

**Plasmids and cell lines.** Sequences of sgRNAs and other oligonucleotides are in Supplementary Table 1. The px330 plasmid was used for sgRNA/Cas9 expression (Addgene 42230, generously deposited by Dr. Feng Zhang)[1], except for the GFPd2-SV assay. Several plasmids were previously described: the EJ7-GFP reporter in the *Pim1* mouse targeting vector, the sgRNA/Cas9 plasmids for the EJ7-GFP reporter (7a and 7b), the sgRNA/Cas9 plasmids for the GAPDH-CD4 assay (GAPDH and CD4), the pCAGGS-3xFLAG-XLF plasmids (both mouse and human), and pCAGGS-HA-XRCC4[29,31]. Expression vectors for mutant forms of 3xFLAG-XLF were either previously described[29,31,52], or generated via cloning in gBLOCKs and/or annealed oligonucleotides (Integrated DNA Technologies). The XLF mutants 6 S/T>A (S132, S203, S245, S251, S263, T266) and 5S/T>D (S203, S245, S251, S263, T266) were inserted into the 3xFLAG-XLF plasmid by PCR amplification from plasmids that were previously described[49], and generously provided by Dr. Mauro Modesti (Cancer Research Center of Marseille). The EJ7-GFP+1 assay uses the 7a +1 sgRNA instead of 7a. The DNAPKcs WT, ABCDE(S/T>A), and PQR(S>A) expression plasmids were previously described (Addgene #83317, #83318 and #83319, respectively, generously deposited by Dr. Katheryn Meek)[54]. As well, the DNAPKcs WT and 898–900>A plasmids were previously described[9], and generously provided by Dr. Katheryn Meek (Michigan State University). The pLKO.1-puro plasmid was used for shRNA experiments (Addgene #8453, generously deposited by Bob Weinberg)[78]. Empty vector (EV) controls for XLF and XRCC4 used pCAGGS-BSKX, and for DNAPKcs used a CMV vector (pCMV6-XL5)[79]. The GFPd2-SV reporter was generated by introducing pCAG-GFPd2 (Addgene #14760, generously deposited by Dr. Connie Cepko)[62] into pCDNA5/FRT (Thermofisher), and the sgRNAs for this reporter were DSB-H, DSB-G, and DSB-L, which were introduced into Piggybac gRNA-puro, which was generously provided by Dr. Allan Bradley (Wellcome Sanger Institute), along with the negative control sgRNA plasmid (#5, No DSB), Piggybac Cas9-Blast plasmid, and Piggybac transposase expression plasmid[60]. The LMNA-HDR assay plasmids (LMNA Cas9/sgRNA, and LMNA-mRuby2-Donor) were previously described, and generously provided by Dr. Jean-Yves Masson (Laval University Cancer Research Center)[59].

Several cell lines were described previously: HEK293 Flp-In T-REx cell lines (EJ7-GFP parental line used to generate *XLF-KO* and *XRCC4-KO*), U2OS (EJ7-GFP parental and EJ7-GFP *XLF-KO*), and mESC EJ7-GFP reporter cell lines (WT, *Xlf−/−*, and *Xrcc4−/−*)[29,31]. The vendor of the parental HEK293 Flp-In T-REx cell line is Invitrogen/Thermofisher, which according to their documentation were derived from The American Type Culture Collection number CRL-1573, which are HEK293, not 293T. The *Prkdc−/−* mESC line was generously provided by Dr. Frederick Alt (Harvard)[80]. Cells were cultured as previously described[29,31], and using the Lonza MycoAlert PLUS Mycoplasma Detection Kit, cell lines tested negative for mycoplasma contamination. Additional mutant cell lines were generated using Cas9/sgRNAs cloned into px330, as described previously[29]. Briefly, Cas9/sgRNA plasmids were co-transfected into cells either with a pgk-puro or dsRED plasmid, transfected cells were enriched using transient puromycin treatment or sorting for dsRED positive cells, followed by plating at low density to isolate and screen individual clones for gene disruption. The *Prkdc−/−* mESC line was used to generate the *Prkdc−/−Xlf−/−* mESC using the sgRNAs mXlfsg1 and mXlfsg2 to create a deletion mutation, and the EJ7-GFP reporter was integrated into these cells by targeting to the *Pim1* locus, as described[31]. The HEK293 EJ7-GFP and U2OS EJ7-GFP cell lines were used to generate the *PRKDC-KO* cell lines with the sgRNAs PRKDCsg1 and PRKDCsg2 to create a deletion mutation, and the HEK293 EJ7-GFP *PRKDC-KO* cell line was used to generate the *PRKDC-KO/XLF-KO* cell line using the sgRNA XLFsg1. For the GFPd2-SV assay, the pCDNA5/FRT-GFPd2 plasmid was introduced into HEK293-FRT cells with pgkFLP as described[79], and then used to generate an *XLF-KO* cell line, using the sgRNA XLFsg1. The 3x-FLAG-XLF-WT and K160D stable expression cell lines were generated by co-transfection of the HEK293 EJ7-GFP *XLF-KO* cell line with these plasmids and pgk-puro, and selection for individual clones with puromycin that were subsequently screened. EV controls for these stable cell lines were generated by co-transfecting pCAGGS-BSKX and pgk-puro, and by pooling puromycin-resistant clones.

**DSB repair reporter assays.** For the EJ7-GFP and EJ7+1-GFP assays with HEK293 and U2OS cells, cells were seeded at $0.5 \times 10^5$ onto a 24 well, and transfected the following day with 200 ng of each sgRNA/Cas9 plasmid (7a and 7b, or 7a+1 and 7b, respectively); 50 ng of XLF expression plasmid, XRCC4 expression plasmid, or control EV; and 200 ng of DNAPKcs expression plasmid or control EV,

using 1.8 µL of Lipofectamine 2000 (Thermofisher) in 0.5 ml of antibiotic-free media. For the LMNA-HDR assay, the LMNA Cas9/sgRNA and LMNA-mRuby2-Donor plasmids replace the two sgRNA/Cas9 plasmids (200 ng each). For mESC EJ7-GFP analysis, the plasmid amounts were 200 ng each Cas9/sgRNA plasmids and 50 ng of XLF expression plasmid, XRCC4 expression plasmid, or EV. Cells were incubated with the transfection reagents for 4 hr, washed, and replaced with complete media, or for the M3814 analysis, with media containing M3814 (i.e., Nedisertib, MedChemExpress #HY-101570 or Selleckchem #S8586) and/or vehicle (Dimethyl Sulfoxide, DMSO) with all wells having the same total amount of DMSO in each experiment. Repair frequencies were normalized to transfection efficiency, which was determined with parallel wells that replace GFP expression vector and EV for the two sgRNA/Cas9 plasmids (i.e., 200 ng pCAGGS-NZE-GFP and 200 ng EV). Cells were analyzed 3 days after transfection using flow cytometry (Dako CyAN ADP, or ACEA Quanteon), as described[31,81]. Summit 4.4 was used for the CyAN, and Agilent NovoExpresss Version 1.5.0 for the Quanteon, with the gating strategies shown in Supplementary Fig. 9. For the GAPDH-CD4 analysis, the transfection conditions were the same, except using the GAPDH and CD4 Cas9/sgRNA plasmids, all amounts were scaled 4-fold to a 6 well dish, and cells stained with phycoerythrin-CD4 antibody (BioLegend, 317410, clone OKT4, 1:500) prior to analysis or isolation of CD4 + cells (BD Aria sorter), as described[29]. For all reporter assays, each bar represents the mean and error bars represent standard deviation, and the number of independent transfections and statistics are as described in the figure legends.

For the GFPd2-SV assay, HEK293-GFPd2-SV cells were transfected as for the other reporter assays, except using the plasmids 200 ng Piggybac transposase, 200 ng Piggybac Cas9-Blast, and 100 ng of the respective Piggybac sgRNA-puro, although many of the experiments were scaled twofold to a 12 well. M3814 treatment was added as for the other reporter assays, and was continual through four days after transfection. The day after transfection, cells were treated with puromycin (2 µg/ml) and blasticidin (7 µg/ml) for a total of 9 days to select for cells with stable expression of sgRNAs/Cas9, and allow for loss of GFPd2 protein. Cells were analyzed for GFP+ frequencies by flow cytometry as for the other reporter assays and/or used for cell sorting of GFP-negative (GFP-neg) cells (BD Aria sorter).

For GFPd2-SV reporter validation, samples sorted to enrich for GFP-neg cells were used for qPCR analysis (BioRad iTaq Universal SYBR Green Supermix #1725120 and Biorad CFX96) with the primers described in Supplementary Table 1. Gemonic DNA was purified as described[81], digested overnight with BglII (New England Biolabs), and column purified (GFX/Illustra) prior to amplification. The Ct values for each locus within the reporter were subtracted from mean values for Actin control ($\Delta Ct$), and then the mean $\Delta Ct$ from the parental reporter cell line was subtracted ($\Delta\Delta Ct$), and used to calculate locus amplification vs. the parental ($2^{-\Delta\Delta Ct}$). To validate the sgRNAs caused DSBs at the predicted site, genomic DNA from transfected cells were amplified with primers flanking each DSB site (Supplementary Table 1), and subjected to Tracking of Indels by Decomposition (i.e., TIDE) analysis[82].

**GAPDH-CD4 rearrangement junction analysis**. CD4 + cells isolated following the DSB reporter assay described above were used to purify genomic DNA, as described[29], and the GAPDH-CD4 rearrangement junction was amplified with ILL-GAPDH and ILL-CD4 primers, which include the Illumina adapter sequences. The amplicons were subjected to deep sequencing using the Amplicon-EZ service (GENEWIZ), which includes their SNP/INDEL detection pipeline that aligned the reads to the No Indel EJ junction sequence as the reference sequence. The percentage of reads for the various indel types (No Indel, deletion, insertion, or complex indel) were quantified for each sample. Each cellular condition was examined with three independent transfections and CD4+ sorted samples, the percentage of each indel type, along with deletion and insertion sizes, from the three samples was used to calculate the mean and standard deviation. Analysis of the insertion sequences was performed on all read sequences representing at least 0.1% of the total insertion reads.

**Clonogenic survival assays**. Cell lines were pre-treated with M3814 or vehicle (DMSO) for 3 h prior to trypsinization and resuspension in the same media as the pre-treatment, counted, and split into two groups: treatment with 0.5 or 1 Gy IR (Gammacell 3000) or left untreated. Cells were then plated in 6 well dishes at various cell densities in the same media as the pre-treatment with M3814 or vehicle (DMSO), and colonies allowed to form for 7–10 days, which were fixed in cold methanol prior to staining with 0.5% crystal violet (Sigma) in 25% methanol. Colonies were counted, with sample identity blinded to experimenter performing the counting, under a 4X objective and clonogenic survival was determined for each well relative to the mean value of DMSO/untreated wells for the respective cell line, with corrections for the plating density.

**Immunoblotting**. Cells were lysed with extraction with ELB (250 mM NaCl, 5 mM EDTA, 50 mM Hepes, 0.1% (v/v) Ipegal, and Roche protease inhibitor) with sonication (Qsonica, Q800R), or NETN Buffer (20 mM TRIS pH 8.0, 100 mM NaCl, 1 mM EDTA, 0.5% IGEPAL, 1.25 mM DTT and Roche Protease Inhibitor) with several freeze/thaw cycles. For DNAPKcs-S2056p analysis, cells were pre-

treated with M3814 or vehicle (DMSO) for 3 h, treated with 10 Gy IR (Gammacell 3000), allowed to recover for 1 h, and protein extracted with ELB buffer containing PhosSTOP (Roche) and 50 µM sodium fluoride. The transfections for immuno-blotting analysis were identical to the reporter assays, except replacing EV (pCAGGS-BSKX) for the sgRNA/Cas9 plasmids, and scaled to a 6 well dish. Blots were probed with antibodies for DNAPKcs (Invitrogen MA5–13238, clone 18-2, 1:1000), DNAPKcs-S2056p (Abcam ab124918, clone EPR5670, 1:1000), XLF (Bethyl A300–730A, 1:1000), XRCC4 (Santa Cruz sc271087, clone C-4, 1:1000), Tubulin (Sigma T9026, clone DM1A, 1:1000), FLAG-HRP (Sigma A8592, clone M2, 1:1000), ACTIN (Sigma A2066), HRP goat anti-mouse (Abcam ab205719, 1:3000), and HRP goat anti-rabbit (Abcam ab205718, 1:3000). ECL reagent (Amersham Biosciences) was used to develop HRP signals.

**Data collection and analysis**. Summit 4.4 with the CyAN, and Agilent NovoExpresss Version 1.5.0 with the Quanteon were used to capture flow cyto-metry data and perform the analysis. Statistical tests were performed with Prism Version 8.3.0. DNAPKcs-S2056p signals were quantified with ImageJ.

**Reporting summary**. Further information on research design is available in the Nature Research Reporting Summary linked to this article.

## Data availability
The datasets generated during and/or analyzed during the current study are included in the study and are available from the corresponding author on reasonable request. Source data are included with this paper. The image of the published structure of the XLF homodimer is from publicly available data from Protein Data Bank 2R9A. Source data are provided with this paper.

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

## Acknowledgements

The authors thank Eva Jahanshir for technical support, and Dr. Jean-Yves Masson and Dr. Amélie Rodrigue (Laval University Cancer Research Center) for technical advice and reagents for the LMNA-HDR assay. This study was funded in part by the National Cancer Institute of the National Institutes of Health: R01CA256989, R01CA197506, R01CA240392 (J.M.S.); P30CA33572 (City of Hope Core Facilities).

## Author contributions

M.C.-A., L.J.T., and J.M.S. designed research and analyzed data; M.C.-A., L.J.T., F.W.L., and R.B. performed research; M.C.-A. and J.M.S. wrote the paper with input from all authors.

## Competing interests

The authors declare no competing interests.

## Additional information

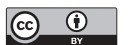

