## [Peer Review File · Nature Communications]

The importance of DNAPKcs for blunt DNA end joining is magnified when XLF is weakenedEditorial Note: This manuscript has been previously reviewed at another journal that is not operating a transparent peer review scheme. This document only contains reviewer comments and rebuttal letters for versions considered at *Nature Communications* .

REVIEWERS' COMMENTS

Reviewer #1 (Remarks to the Author):

Response to rebuttal:

1. The first comment requested a more thorough exploration of mutations affecting the XLF homodimer interface in order to strengthen the argument that the observed effects caused by the K160D mutation were the result of a weakened dimer interface rather than variations in XLF expression. In response, the authors tested the suggested K160A and K160D/D161K mutants, finding the former had a similar but milder phenotype to their original mutation and supporting their original conclusion. Furthermore, the authors have made improvements to the main text, clarifying their investigation of other mutants, and the double mutant XLF-L115A/4KA is added, providing convincing evidence that the weakening the XLF dimer interface is the major factor driving the observed decrease in No Indel EJ.

To further address the affects XLF expression levels, the reviewers performed several shRA-depletion and overexpression experiments, observing dose dependent effects on No Indel EJ only when DNA-PKcs is knocked out. This is a strong addition to the manuscript, and it reinforces their arguments.

2. The second comment notes that XLF-XRCC4 overexpression in mESC increases No Indel EJ to levels greater than WT, while HEK293 fail to reach WT levels under the same overexpression conditions. In their response, the authors hypothesize XLF-XRCC4 levels are not rate limiting in HEK293 cells and test this by overexpressing the proteins in the parental cell line. Observing no increase in No Indel EJ under these conditions, the authors provide sufficient evidence for the differences between cell lines.

The second comment also noted a small point about DNA-PKcs expression, which the authors clarified in the main text.

3. Comment three concerns how fold-effects are presented. The authors have made changes to the text clarifying their rationale, which is to allow the reader to assess the relationship between different disruptions. Furthermore, the fold-effects in M3814 +/- DNA-PKcs loss are now compared with respect to the XLF mutant, which is a more meaningful comparison. These changes improve these figures and make the data more easily interpretable.

4. Here a more thorough exploration of the specific effects of M3814 was requested because they could not detect dose-dependent changes PQR phosphorylation, while No Indel EJ showed dose-dependency. The authors provide an experiment which attempted to test phosphorylation of the ABCDE cluster, detection using the an antibody for T2609p was unsuccessful. Absent direct evidence as to how M3814 inhibits DNA-PKcs activity, the authors provide enough additional evidence to argue DNA-PKcs inhibition is still responsible for the dose-dependency via interrogation of PIKK targets and noting that M3814 does not affect No Indel EJ in DNA-PKcs knockout cells. Alternative techniques to further investigate M3814 effects different DNA-PKcs phosphorylation clusters could be proposed, but an exhaustive investigation is likely beyond the purview of the manuscript.

5. Comment five requests a more information and a more thorough discussion of the variety of overhangs produced in the GAPDH-CD4 assay. The authors provide sufficient new analysis in the main text, figure 4, and supplementary figure, to address this.

6. The reviewers provide a robust justification for their use of only 5' overhangs. Clarification has been added to the main text that the discussion of overhangs is limited to staggered Cas9 cutting rather than more complicated models of DSBs that produce other varieties of DNA ends.

Minor concerns:

Typo in supplementary figure 2; 'parformaldehyde' should be 'paraformaldehyde'

Other minor concerns have been addressed.

Reviewer #2 (Remarks to the Author):

The authors' response and revisions have satisfactorily addressed the reviewers' comments on the earlier version of the manuscript. Publication of this version is recommended.

Reviewer #3 (Remarks to the Author):

I acknowledge the effort of the authors in addressing my concerns together with that of the other reviewers and in producing substantial amount of control experiments and new data that significantly add to the quality of the study, strengthen and broaden the conclusions.

I take the opportunity of reviewing this revised version to make some suggestions

- given that the scope of the study is to examine redundancies between XLF and DNA-PKcs especially regarding the synapsis, XRCC4 effect on EJ should be removed from the abstract since this not adds to the message

- p10, the control of nuclear localization of WT and mutant XLF (line 192) may rather be shifted to the next paragraph (line 202) dealing with similar control experiments related to XLF, rather than be attached to experiments with M3814 DNA-PKi

- I believe that the results of altering the levels of XLF on EJ add to the model since they reveal an effect only under DNA-PKcs loss. Therefore, this could be somehow emphasized in Figure 7, second case, by pointing out a limiting effect of XLF under PRKDC-KO conditions.

- p19, line 415, the figure referred to is Supplementary Fig. 6c and not 5c

- I doubt that EJ of blunt DSBs actually models the final ligation step of c-NHEJ as stated in the discussion (p26 line 550), otherwise conclusions of data with the blunt EJ reporter would be identical to that of IR-sensitivity data, that is not the case. This should be discussed or sensitivity to zeocin that more specifically produces blunt DSBs (10.1021/bi00440a016) should be assessed rather than to IR.

Point-by-point responses to the reviewers' comments for the revised manuscript of NCOMMS-22-02308A. We thank the reviewers for their suggestions to improve the manuscript. We have responded to each concern with the requested edits to the text.

REVIEWERS' COMMENTS

Reviewer #1 (Remarks to the Author):

Response to rebuttal:

1. The first comment requested a more thorough exploration of mutations affecting the XLF homodimer interface in order to strengthen the argument that the observed effects caused by the K160D mutation were the result of a weakened dimer interface rather than variations in XLF expression. In response, the authors tested the suggested K160A and K160D/D161K mutants, finding the former had a similar but milder phenotype to their original mutation and supporting their original conclusion. Furthermore, the authors have made improvements to the main text, clarifying their investigation of other mutants, and the double mutant XLF-L115A/4KA is added, providing convincing evidence that the weakening the XLF dimer interface is the major factor driving the observed decrease in No Indel EJ.

To further address the affects XLF expression levels, the reviewers performed several shRA-depletion and overexpression experiments, observing dose dependent effects on No Indel EJ only when DNA-PKcs is knocked out. This is a strong addition to the manuscript, and it reinforces their arguments.

2. The second comment notes that XLF-XRCC4 overexpression in mESC increases No Indel EJ to levels greater than WT, while HEK293 fail to reach WT levels under the same overexpression conditions. In their response, the authors hypothesize XLF-XRCC4 levels are not rate limiting in HEK293 cells and test this by overexpressing the proteins in the parental cell line. Observing no increase in No Indel EJ under these conditions, the authors provide sufficient evidence for the differences between cell lines.

The second comment also noted a small point about DNA-PKcs expression, which the authors clarified in the main text.

3. Comment three concerns how fold-effects are presented. The authors have made changes to the text clarifying their rationale, which is to allow the reader to assess the relationship between different disruptions. Furthermore, the fold-effects in M3814 +/- DNA-PKcs loss are now compared with respect to the XLF mutant, which is a more meaningful comparison. These changes improve these figures and make the data more easily interpretable.

4. Here a more thorough exploration of the specific effects of M3814 was requested because they could not detect dose-dependent changes PQR phosphorylation, while No Indel EJ showed dose-dependency. The authors provide an experiment which attempted to test phosphorylation of the ABCDE cluster, detection using the an antibody for T2609p was unsuccessful. Absent direct evidence as to how M3814 inhibits DNA-PKcs activity, the authors provide enough additional evidence to argue DNA-PKcs inhibition is still responsible for the dose-dependency via interrogation of PIKK targets and noting that M3814 does not affect No Indel EJ in DNA-PKcs knockout cells. Alternative techniques to further investigate M3814 effects different DNA-PKcs phosphorylation clusters could be proposed, but an

exhaustive investigation is likely beyond the purview of the manuscript.

5. Comment five requests a more information and a more thorough discussion of the variety of overhangs produced in the GAPDH-CD4 assay. The authors provide sufficient new analysis in the main text, figure 4, and supplementary figure, to address this.

6. The reviewers provide a robust justification for their use of only 5' overhangs. Clarification has been added to the main text that the discussion of overhangs is limited to staggered Cas9 cutting rather than more complicated models of DSBs that produce other varieties of DNA ends.

The reviewer has provided a detailed accounting of our responses to their prior critique, and no new concerns have been raised.

Minor concerns:

Typo in supplementary figure 2; 'parformaldehyde' should be 'paraformaldehyde'

Point 1: The reviewer has noted a typographical error that we have fixed.

Other minor concerns have been addressed.

Reviewer #2 (Remarks to the Author):

The authors' response and revisions have satisfactorily addressed the reviewers' comments on the earlier version of the manuscript. Publication of this version is recommended.

The reviewer has not raised any concerns.

Reviewer #3 (Remarks to the Author):

I acknowledge the effort of the authors in addressing my concerns together with that of the other reviewers and in producing substantial amount of control experiments and new data that significantly add to the quality of the study, strengthen and broaden the conclusions.

I take the opportunity of reviewing this revised version to make some suggestions

- given that the scope of the study is to examine redundancies between XLF and DNA-PKcs especially regarding the synapsis, XRCC4 effect on EJ should be removed from the abstract since this not adds to the message

Point 1: The reviewer requests that we remove "XRCC4" from the Abstract, and we have performed the requested edit.

- p10, the control of nuclear localization of WT and mutant XLF (line 192) may rather be shifted to the next paragraph (line 202) dealing with similar control experiments related to XLF, rather than be attached to experiments with M3814 DNA-PKi

Point 2: The reviewer requests that the immunofluorescence analysis sentence regarding XLF WT and K160D be moved later in the Results, so that the M3814 controls are grouped together. We have performed the requested edit in this paragraph, which lists a series of controls. Specifically, we have grouped the M3814 controls at the beginning of the paragraph (current Supplementary Figure 1d, 2a), and then placed the immunofluorescence analysis sentence at the end of the paragraph (Supplementary Figure 2b). Accordingly, this edit has also resulted in reversing the order of Supplementary Figure 2a and 2b.

- I believe that the results of altering the levels of XLF on EJ add to the model since they reveal an effect only under DNA-PKcs loss. Therefore, this could be somehow emphasized in Figure 7, second case, by pointing out a limiting effect of XLF under PRKDC-KO conditions.

Point 3: The reviewer recommends that we add “reduced levels of XLF” to the part of Figure 7 (the model figure) relating to combinations with DNAPKcs loss (PRKDC-KO), and we have performed the requested edit to the figure.

- p19, line 415, the figure referred to is Supplementary Fig. 6c and not 5c

Point 4: The reviewer identified a typo for a figure callout, which we have fixed.

- I doubt that EJ of blunt DSBs actually models the final ligation step of c-NHEJ as stated in the discussion (p26 line 550), otherwise conclusions of data with the blunt EJ reporter would be identical to that of IR-sensitivity data, that is not the case. This should be discussed or sensitivity to zeocin that more specifically produces blunt DSBs (10.1021/bi00440a016) should be assessed rather than to IR.

Point 5: The reviewer disagrees with our assertion in the Discussion section that end joining (EJ) of blunt DNA double-strand breaks (DSBs) models the “final ligation step of C-NHEJ after end processing.” In response, we have removed this phrase from this sentence, which accordingly makes the sentence more conservative/cautious. Specifically, we have made the following edit:

Original sentence: “Furthermore, we have examined EJ of blunt DSBs to model the final ligation step of C-NHEJ after end processing to yield ligatable ends that may not be stabilized by an annealing intermediate”

Revised sentence: “Furthermore, we have examined EJ of blunt DSBs to model repair of ligatable ends that may not be stabilized by an annealing intermediate”